# Provable Non-convex Robust PCA

**Praneeth Netrapalli** [1]*    **U N Niranjan**[2]*    **Sujay Sanghavi**[3]    **Animashree Anandkumar**[2]

**Prateek Jain**[4]

[1]Microsoft Research, Cambridge MA. [2]The University of California at Irvine.
[3]The University of Texas at Austin. [4]Microsoft Research, India.

## Abstract

We propose a new method for robust PCA – the task of recovering a low-rank matrix from sparse corruptions that are of unknown value and support. Our method involves alternating between projecting appropriate residuals onto the set of low-rank matrices, and the set of sparse matrices; each projection is *non-convex* but easy to compute. In spite of this non-convexity, we establish exact recovery of the low-rank matrix, under the same conditions that are required by existing methods (which are based on convex optimization). For an $m \times n$ input matrix ($m \leq n$), our method has a running time of $O\left(r^2 mn\right)$ per iteration, and needs $O\left(\log(1/\epsilon)\right)$ iterations to reach an accuracy of $\epsilon$. This is close to the running times of simple PCA via the power method, which requires $O\left(rmn\right)$ per iteration, and $O\left(\log(1/\epsilon)\right)$ iterations. In contrast, the existing methods for robust PCA, which are based on convex optimization, have $O\left(m^2 n\right)$ complexity per iteration, and take $O\left(1/\epsilon\right)$ iterations, i.e., exponentially more iterations for the same accuracy.

Experiments on both synthetic and real data establishes the improved speed and accuracy of our method over existing convex implementations.

**Keywords:**    Robust PCA, matrix decomposition, non-convex methods, alternating projections.

## 1 Introduction

Principal component analysis (PCA) is a common procedure for preprocessing and denoising, where a low rank approximation to the input matrix (such as the covariance matrix) is carried out. Although PCA is simple to implement via eigen-decomposition, it is sensitive to the presence of outliers, since it attempts to "force fit" the outliers to the low rank approximation. To overcome this, the notion of robust PCA is employed, where the goal is to remove sparse corruptions from an input matrix and obtain a low rank approximation. Robust PCA has been employed in a wide range of applications, including background modeling [LHGT04], 3d reconstruction [MZYM11], robust topic modeling [Shi13], and community detection [CSX12], and so on.

Concretely, robust PCA refers to the following problem: given an input matrix $M = L^* + S^*$, the goal is to decompose it into sparse $S^*$ and low rank $L^*$ matrices. The seminal works of [CSPW11, CLMW11] showed that this problem can be provably solved via convex relaxation methods, under some natural conditions on the low rank and sparse components. While the theory is elegant, in practice, convex techniques are expensive to run on a large scale and have poor convergence rates. Concretely, for decomposing an $m \times n$ matrix, say with $m \leq n$, the best specialized implementations (typically first-order methods) have a *per-iteration complexity* of $O\left(m^2 n\right)$, and require $O(1/\epsilon)$ number of iterations to achieve an error of $\epsilon$. In contrast, the usual PCA, which carries out a rank-$r$ approximation of the input matrix, has $O(rmn)$ complexity per iteration – drastically smaller

when $r$ is much smaller than $m, n$. Moreover, PCA requires exponentially fewer iterations for convergence: an $\epsilon$ accuracy is achieved with only $O\left(\log(1/\epsilon)\right)$ iterations (assuming constant gap in singular values).

In this paper, we design a non-convex algorithm which is "best of both the worlds" and bridges the gap between (the usual) PCA and convex methods for robust PCA. Our method has low computational complexity similar to PCA (i.e. scaling costs and convergence rates), and at the same time, has provable global convergence guarantees, similar to the convex methods. Proving global convergence for non-convex methods is an exciting recent development in machine learning. Non-convex alternating minimization techniques have recently shown success in many settings such as matrix completion [Kes12, JNS13, Har13], phase retrieval [NJS13], dictionary learning [AAJ+13], tensor decompositions for unsupervised learning [AGH+12], and so on. Our current work on the analysis of non-convex methods for robust PCA is an important addition to this growing list.

## 1.1 Summary of Contributions

We propose a simple intuitive algorithm for robust PCA with low per-iteration cost and a fast convergence rate. We prove tight guarantees for recovery of sparse and low rank components, which match those for the convex methods. In the process, we derive novel matrix perturbation bounds, when subject to sparse perturbations. Our experiments reveal significant gains in terms of speed-ups over the convex relaxation techniques, especially as we scale the size of the input matrices.

Our method consists of simple alternating (non-convex) projections onto low-rank and sparse matrices. For an $m \times n$ matrix, our method has a running time of $O(r^2 mn \log(1/\epsilon))$, where $r$ is the rank of the low rank component. Thus, our method has a linear convergence rate, i.e. it requires $O(\log(1/\epsilon))$ iterations to achieve an error of $\epsilon$, where $r$ is the rank of the low rank component $L^*$. When the rank $r$ is small, this nearly matches the complexity of PCA, (which is $O(rmn \log(1/\epsilon))$).

We prove recovery of the sparse and low rank components under a set of requirements which are tight and match those for the convex techniques (up to constant factors). In particular, under the deterministic sparsity model, where each row and each column of the sparse matrix $S^*$ has at most $\alpha$ fraction of non-zeros, we require that $\alpha = O\left(1/(\mu^2 r)\right)$, where $\mu$ is the incoherence factor (see Section 3).

In addition to strong theoretical guarantees, in practice, our method enjoys significant advantages over the state-of-art solver for (1), viz., the inexact augmented Lagrange multiplier (IALM) method [CLMW11]. Our method outperforms IALM in all instances, as we vary the sparsity levels, incoherence, and rank, in terms of running time to achieve a fixed level of accuracy. In addition, on a real dataset involving the standard task of foreground-background separation [CLMW11], our method is significantly faster and provides visually better separation.

**Overview of our techniques:** Our proof technique involves establishing error contraction with each projection onto the sets of low rank and sparse matrices. We first describe the proof ideas when $L^*$ is rank one. The first projection step is a hard thresholding procedure on the input matrix $M$ to remove large entries and then we perform rank-1 projection of the residual to obtain $L^{(1)}$. Standard matrix perturbation results (such as Davis-Kahan) provide $\ell_2$ error bounds between the singular vectors of $L^{(1)}$ and $L^*$. However, these bounds do not suffice for establishing the correctness of our method. Since the next step in our method involves hard thresholding of the residual $M - L^{(1)}$, we require element-wise error bounds on our low rank estimate. Inspired by the approach of Erdős et al. [EKYY13], where they obtain similar element-wise bounds for the eigenvectors of sparse Erdős–Rényi graphs, we derive these bounds by exploiting the fixed point characterization of the eigenvectors[1]. A Taylor's series expansion reveals that the perturbation between the estimated and the true eigenvectors consists of bounding the walks in a graph whose adjacency matrix corresponds to (a subgraph of) the sparse component $S^*$. We then show that if the graph is sparse enough, then this perturbation can be controlled, and thus, the next thresholding step results in further error contraction. We use an induction argument to show that the sparse estimate is always contained in the true support of $S^*$, and that there is an error contraction in each step. For the case, where $L^*$ has rank $r > 1$, our algorithm proceeds in several stages, where we progressively compute higher rank

projections which alternate with the hard thresholding steps. In stage $k = [1, 2, \ldots, r]$, we compute rank-$k$ projections, and show that after a sufficient number of alternating projections, we reduce the error to the level of $(k + 1)^{\text{th}}$ singular value of $L^*$, using similar arguments as in the rank-1 case. We then proceed to performing rank-$(k + 1)$ projections which alternate with hard thresholding. This stage-wise procedure is needed for ill-conditioned matrices, since we cannot hope to recover lower eigenvectors in the beginning when there are large perturbations. Thus, we establish global convergence guarantees for our proposed non-convex robust PCA method.

## 1.2 Related Work

Guaranteed methods for robust PCA have received a lot of attention in the past few years, starting from the seminal works of [CSPW11, CLMW11], where they showed recovery of an incoherent low rank matrix $L^*$ through the following convex relaxation method:

$$\text{Conv-RPCA}: \qquad \min_{L,S} \|L\|_* + \lambda\|S\|_1, \qquad \text{s.t.,} \quad M = L + S, \tag{1}$$

where $\|L\|_*$ denotes the nuclear norm of $L$ (nuclear norm is the sum of singular values). A typical solver for this convex program involves projection on to $\ell_1$ and nuclear norm balls (which are convex sets). Note that the convex method can be viewed as "soft" thresholding in the standard and spectral domains, while our method involves hard thresholding in these domains.

[CSPW11] and [CLMW11] consider two different models of sparsity for $S^*$. Chandrasekaran et al. [CSPW11] consider a deterministic sparsity model, where each row and column of the $m \times n$ matrix, $S$, has at most $\alpha$ fraction of non-zero entries. For guaranteed recovery, they require $\alpha = O\left(1/(\mu^2 r \sqrt{n})\right)$, where $\mu$ is the incoherence level of $L^*$, and $r$ is its rank. Hsu et al. [HKZ11] improve upon this result to obtain guarantees for an optimal sparsity level of $\alpha = O\left(1/(\mu^2 r)\right)$. This *matches* the requirements of our non-convex method for exact recovery. Note that when the rank $r = O(1)$, this allows for a constant fraction of corrupted entries. Candès et al. [CLMW11] consider a different model with random sparsity and additional incoherence constraints, viz., they require $\|UV^\top\|_\infty < \mu\sqrt{r}/n$. Note that our assumption of incoherence, viz., $\|U^{(i)}\| < \mu\sqrt{r/n}$, only yields $\|UV^\top\|_\infty < \mu^2 r/n$. The additional assumption enables [CLMW11] to prove exact recovery with a constant fraction of corrupted entries, even when $L^*$ is nearly full-rank. We note that removing the $\|UV^\top\|_\infty$ condition for robust PCA would imply solving the planted clique problem when the clique size is less than $\sqrt{n}$ [Che13]. Thus, our recovery guarantees are *tight* upto constants without these additional assumptions.

A number of works have considered modified models under the robust PCA framework, e.g. [ANW12, XCS12]. For instance, Agarwal et al. [ANW12] relax the incoherence assumption to a weaker "diffusivity" assumption, which bounds the magnitude of the entries in the low rank part, but incurs an additional approximation error. Xu et al.[XCS12] impose special sparsity structure where a column can either be non-zero or fully zero.

In terms of state-of-art specialized solvers, [CLMW11] implements the in-exact augmented Lagrangian multipliers (IALM) method and provides guidelines for parameter tuning. Other related methods such as multi-block alternating directions method of multipliers (ADMM) have also been considered for robust PCA, e.g. [WHML13]. Recently, a multi-step multi-block stochastic ADMM method was analyzed for this problem [SAJ14], and this requires $1/\epsilon$ iterations to achieve an error of $\epsilon$. In addition, the convergence rate is tight in terms of scaling with respect to problem size $(m, n)$ and sparsity and rank parameters, under random noise models.

There is only one other work which considers a non-convex method for robust PCA [KC12]. However, their result holds only for significantly more restrictive settings and does not cover the deterministic sparsity assumption that we study. Moreover, the projection step in their method can have an arbitrarily large rank, so the running time is still $O(m^2 n)$, which is the same as the convex methods. In contrast, we have an improved running time of $O(r^2 mn)$.

## 2 Algorithm

In this section, we present our algorithm for the robust PCA problem. The robust PCA problem can be formulated as the following optimization problem: find $L, S$ s.t. $\|M - L - S\|_F \leq \epsilon^2$ and

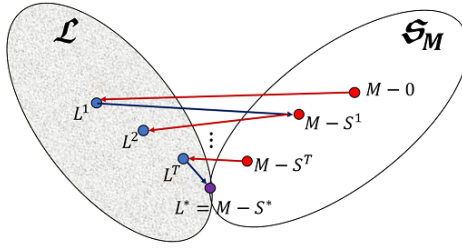

Figure 1: Illustration of alternating projections. The goal is to find a matrix $L^*$ which lies in the intersection of two sets: $\mathcal{L} = \{$ set of rank-$r$ matrices$\}$ and $\mathcal{S}_M = \{M - S, \text{ where } S \text{ is a sparse matrix}\}$. Intuitively, our algorithm alternately projects onto the above two non-convex sets, while appropriately relaxing the rank and the sparsity levels.

1. $L$ lies in the set of low-rank matrices,

2. $S$ lies in the set of sparse matrices.

A natural algorithm for the above problem is to iteratively project $M - L$ onto the set of sparse matrices to update $S$, and then to project $M - S$ onto the set of low-rank matrices to update $L$. Alternatively, one can view the problem as that of finding a matrix $L$ in the intersection of the following two sets: a) $\mathcal{L} = \{$ set of rank-$r$ matrices$\}$, b) $\mathcal{S}_M = \{M - S, \text{ where } S \text{ is a sparse matrix}\}$. Note that these projections can be done efficiently, even though the sets are non-convex. Hard thresholding (HT) is employed for projections on to sparse matrices, and singular value decomposition (SVD) is used for projections on to low rank matrices.

**Rank-1 case:** We first describe our algorithm for the special case when $L^*$ is rank 1. Our algorithm performs an initial hard thresholding to remove very large entries from input $M$. Note that if we performed the projection on to rank-1 matrices without the initial hard thresholding, we would not make any progress since it is subject to large perturbations. We alternate between computing the rank-1 projection of $M - S$, and performing hard thresholding on $M - L$ to remove entries exceeding a certain threshold. This threshold is gradually decreased as the iterations proceed, and the algorithm is run for a certain number of iterations (which depends on the desired reconstruction error).

**General rank case:** When $L^*$ has rank $r > 1$, a naive extension of our algorithm consists of alternating projections on to rank-$r$ matrices and sparse matrices. However, such a method has poor performance on ill-conditioned matrices. This is because after the initial thresholding of the input matrix $M$, the sparse corruptions in the residual are of the order of the top singular value (with the choice of threshold as specified in the algorithm). When the lower singular values are much smaller, the corresponding singular vectors are subject to relatively large perturbations and thus, we cannot make progress in improving the reconstruction error. To alleviate the dependence on the condition number, we propose an algorithm that proceeds in stages. In the $k^{\text{th}}$ stage, the algorithm alternates between rank-$k$ projections and hard thresholding for a certain number of iterations. We run the algorithm for $r$ stages, where $r$ is the rank of $L^*$. Intuitively, through this procedure, we recover the lower singular values only after the input matrix is sufficiently denoised, i.e. sparse corruptions at the desired level have been removed. Figure 1 shows a pictorial representation of the alternating projections in different stages.

**Parameters:** As can be seen, the only real parameter to the algorithm is $\beta$, used in thresholding, which represents "spikiness" of $L^*$. That is if the user expects $L^*$ to be "spiky" and the sparse part to be heavily diffused, then higher value of $\beta$ can be provided. In our implementation, we found that selecting $\beta$ aggressively helped speed up recovery of our algorithm. In particular, we selected $\beta = 1/\sqrt{n}$.

**Complexity:** The complexity of each iteration within a single stage is $O(kmn)$, since it involves calculating the rank-$k$ approximation[3] of an $m \times n$ matrix (done e.g. via vanilla PCA). The number of iterations in each stage is $O\left(\log\left(1/\epsilon\right)\right)$ and there are at most $r$ stages. Thus the overall complexity of the entire algorithm is then $O(r^2 mn \log(1/\epsilon))$. This is drastically lower than the best known bound of $O\left(m^2 n/\epsilon\right)$ on the number of iterations required by convex methods, and just a factor $r$ away from the complexity of vanilla PCA.

**Algorithm 1** $(\widehat{L}, \widehat{S}) = \text{AltProj}(M, \epsilon, r, \beta)$: Non-convex Alternating Projections based Robust PCA

1: **Input**: Matrix $M \in \mathbb{R}^{m \times n}$, convergence criterion $\epsilon$, target rank $r$, thresholding parameter $\beta$.
2: $P_k(A)$ denotes the best rank-$k$ approximation of matrix $A$. $HT_\zeta(A)$ denotes hard-thresholding, i.e. $(HT_\zeta(A))_{ij} = A_{ij}$ if $|A_{ij}| \geq \zeta$ and 0 otherwise.
3: Set initial threshold $\zeta_0 \leftarrow \beta\sigma_1(M)$.
4: $L^{(0)} = 0, S^{(0)} = HT_{\zeta_0}(M - L^{(0)})$
5: **for** Stage $k = 1$ to $r$ **do**
6:     **for** Iteration $t = 0$ to $T = 10\log\left(n\beta \left\|M - S^{(0)}\right\|_2 /\epsilon\right)$ **do**
7:         Set threshold $\zeta$ as

$$\zeta = \beta\left(\sigma_{k+1}(M - S^{(t)}) + \left(\frac{1}{2}\right)^t \sigma_k(M - S^{(t)})\right) \quad (2)$$

8:         $L^{(t+1)} = P_k(M - S^{(t)})$
9:         $S^{(t+1)} = HT_\zeta(M - L^{(t+1)})$
10:     **end for**
11:     **if** $\beta\sigma_{k+1}(L^{(t+1)}) < \frac{\epsilon}{2n}$ **then**
12:         **Return:** $L^{(T)}, S^{(T)}$   */\* Return rank-$k$ estimate if remaining part has small norm \*/*
13:     **else**
14:         $S^{(0)} = S^{(T)}$                */\* Continue to the next stage \*/*
15:     **end if**
16: **end for**
17: **Return:** $L^{(T)}, S^{(T)}$

## 3 Analysis

In this section, we present our main result on the correctness of AltProj. We assume the following conditions:

(L1) Rank of $L^*$ is at most $r$.

(L2) $L^*$ is $\mu$-incoherent, i.e., if $L^* = U^*\Sigma^*(V^*)^\top$ is the SVD of $L^*$, then $\|(U^*)^i\|_2 \leq \frac{\mu\sqrt{r}}{\sqrt{m}}$, $\forall 1 \leq i \leq m$ and $\|(V^*)^i\|_2 \leq \frac{\mu\sqrt{r}}{\sqrt{n}}$, $\forall 1 \leq i \leq n$, where $(U^*)^i$ and $(V^*)^i$ denote the $i^{\text{th}}$ rows of $U^*$ and $V^*$ respectively.

(S1) Each row and column of $S$ have at most $\alpha$ fraction of non-zero entries such that $\alpha \leq \frac{1}{512\mu^2 r}$.

Note that in general, it is not possible to have a unique recovery of low-rank and sparse components. For example, if the input matrix $M$ is both sparse and low rank, then there is no unique decomposition (e.g. $M = e_1 e_1^\top$). The above conditions ensure uniqueness of the matrix decomposition problem.

Additionally, we set the parameter $\beta$ in Algorithm 1 be set as $\beta = \frac{4\mu^2 r}{\sqrt{mn}}$.

We now establish that our proposed algorithm recovers the low rank and sparse components under the above conditions.

**Theorem 1** (Noiseless Recovery). *Under conditions* (L1)*,* (L2) *and* $S^*$*, and choice of* $\beta$ *as above, the outputs* $\widehat{L}$ *and* $\widehat{S}$ *of Algorithm 1 satisfy:*

$$\left\|\widehat{L} - L^*\right\|_F \leq \epsilon, \left\|\widehat{S} - S^*\right\|_\infty \leq \frac{\epsilon}{\sqrt{mn}}, \text{ and } \text{Supp}\left(\widehat{S}\right) \subseteq \text{Supp}\left(S^*\right).$$

**Remark (tight recovery conditions):** Our result is tight up to constants, in terms of allowable sparsity level under the deterministic sparsity model. In other words, if we exceed the sparsity limit imposed in S1, it is possible to construct instances where there is no unique decomposition[4]. Our

conditions L1, L2 and S1 also match the conditions required by the convex method for recovery, as established in [HKZ11].

**Remark (convergence rate):** Our method has a linear rate of convergence, i.e. $O(\log(1/\epsilon))$ to achieve an error of $\epsilon$, and hence we provide a strongly polynomial method for robust PCA. In contrast, the best known bound for convex methods for robust PCA is $O(1/\epsilon)$ iterations to converge to an $\epsilon$-approximate solution.

Theorem 1 provides recovery guarantees assuming that $L^*$ is exactly rank-$r$. However, in several real-world scenarios, $L^*$ can be nearly rank-$r$. Our algorithm can handle such situations, where $M = L^* + N^* + S^*$, with $N^*$ being an additive noise. Theorem 1 is a special case of the following theorem which provides recovery guarantees when $N^*$ has small $\ell_\infty$ norm.

**Theorem 2** (Noisy Recovery). *Under conditions* $(L1)$, $(L2)$ *and* $S^*$, *and choice of* $\beta$ *as in Theorem 1, when the noise* $\|N^*\|_\infty \leq \frac{\sigma_r(L^*)}{100n}$, *the outputs* $\widehat{L}, \widehat{S}$ *of Algorithm 1 satisfy:*

$$\left\|\widehat{L} - L^*\right\|_F \leq \epsilon + 2\mu^2 r \left( 7 \|N^*\|_2 + \frac{8\sqrt{mn}}{\sqrt{r}} \|N^*\|_\infty \right),$$

$$\left\|\widehat{S} - S^*\right\|_\infty \leq \frac{\epsilon}{\sqrt{mn}} + \frac{2\mu^2 r}{\sqrt{mn}} \left( 7 \|N^*\|_2 + \frac{8\sqrt{mn}}{\sqrt{r}} \|N^*\|_\infty \right), \text{ and } Supp\left(\widehat{S}\right) \subseteq Supp\left(S^*\right).$$

## 3.1 Proof Sketch

We now present the key steps in the proof of Theorem 1. A detailed proof is provided in the appendix.

**Step I: Reduce to the symmetric case, while maintaining incoherence of $L^*$ and sparsity of $S^*$.**
Using standard symmetrization arguments, we can reduce the problem to the symmetric case, where all the matrices involved are symmetric. See appendix for details on this step.

**Step II: Show decay in $\|L - L^*\|_\infty$ after projection onto the set of rank-$k$ matrices.** The $t$-th iterate $L^{(t+1)}$ of the $k$-th stage is given by $L^{(t+1)} = P_k(L^* + S^* - S^{(t)})$. Hence, $L^{(t+1)}$ is obtained by using the top principal components of a perturbation of $L^*$ given by $L^* + (S^* - S^{(t)})$. The key step in our analysis is to show that when an incoherent and low-rank $L^*$ is perturbed by a sparse matrix $S^* - S^{(t)}$, then $\|L^{(t+1)} - L^*\|_\infty$ is small and is much smaller than $|S^* - S^{(t)}|_\infty$. The following lemma formalizes the intuition; see the appendix for a detailed proof.

**Lemma 1.** *Let $L^*, S^*$ be symmetric and satisfy the assumptions of Theorem 1 and let $S^{(t)}$ and $L^{(t)}$ be the $t^{th}$ iterates of the $k^{th}$ stage of Algorithm 1. Let $\sigma_1^*, \ldots, \sigma_n^*$ be the eigenvalues of $L^*$, s.t., $|\sigma_1^*| \geq \cdots \geq |\sigma_r^*|$. Then, the following holds:*

$$\left\|L^{(t+1)} - L^*\right\|_\infty \leq \frac{2\mu^2 r}{n} \left( |\sigma_{k+1}^*| + \left(\frac{1}{2}\right)^t |\sigma_k^*| \right),$$

$$\left\|S^* - S^{(t+1)}\right\|_\infty \leq \frac{8\mu^2 r}{n} \left( |\sigma_{k+1}^*| + \left(\frac{1}{2}\right)^t |\sigma_k^*| \right), \text{ and } Supp\left(S^{(t+1)}\right) \subseteq Supp\left(S^*\right).$$

*Moreover, the outputs $\widehat{L}$ and $\widehat{S}$ of Algorithm 1 satisfy:*

$$\left\|\widehat{L} - L^*\right\|_F \leq \epsilon, \quad \left\|\widehat{S} - S^*\right\|_\infty \leq \frac{\epsilon}{n}, \text{ and } Supp\left(\widehat{S}\right) \subseteq Supp\left(S^*\right).$$

**Step III: Show decay in $\|S - S^*\|_\infty$ after projection onto the set of sparse matrices.** We next show that if $\|L^{(t+1)} - L^*\|_\infty$ is much smaller than $\|S^{(t)} - S^*\|_\infty$ then the iterate $S^{(t+1)}$ also has a much smaller error (w.r.t. $S^*$) than $S^{(t)}$. The above given lemma formally provides the error bound.

**Step IV: Recurse the argument.** We have now reduced the $\ell_\infty$ norm of the sparse part by a factor of half, while maintaining its sparsity. We can now go back to steps II and III and repeat the arguments for subsequent iterations.

---

a fraction of $\alpha = O(1/r)$ sparse perturbations suffice to erase one of these blocks making it impossible to recover the matrix.

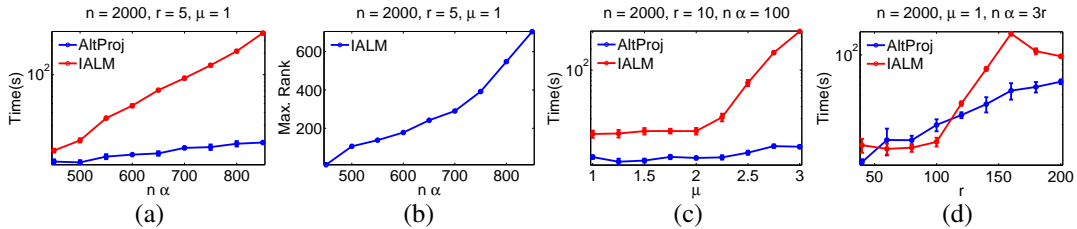

Figure 2: Comparison of AltProj and IALM on synthetic datasets. (a) Running time of AltProj and IALM with varying $\alpha$. (b) Maximum rank of the intermediate iterates of IALM. (c) Running time of AltProj and IALM with varying $\mu$. (d) Running time of AltProj and IALM with varying $r$.

## 4 Experiments

We now present an empirical study of our AltProj method. The goal of this study is two-fold: a) establish that our method indeed recovers the low-rank and sparse part exactly, without significant parameter tuning, b) demonstrate that AltProj is significantly faster than Conv-RPCA (see (1)); we solve Conv-RPCA using the IALM method [CLMW11], a state-of-the-art solver [LCM10]. We implemented our method in Matlab and used a Matlab implementation of the IALM method by [LCM10].

We consider both synthetic experiments and experiments on real data involving the problem of foreground-background separation in a video. Each of our results for synthetic datasets is averaged over $5$ runs.

*Parameter Setting*: Our pseudo-code (Algorithm 1) prescribes the threshold $\zeta$ in Step 4, which depends on the knowledge of the singular values of the low rank component $L^*$. Instead, in the experiments, we set the threshold at the $(t+1)$-th step of $k$-th stage as $\zeta = \frac{\mu\sigma_{k+1}(M-S^{(t)})}{\sqrt{n}}$. For synthetic experiments, we employ the $\mu$ used for data generation, and for real-world datasets, we tune $\mu$ through cross-validation. We found that the above thresholding provides exact recovery while speeding up the computation significantly. We would also like to note that [CLMW11] sets the regularization parameter $\lambda$ in Conv-RPCA (1) as $1/\sqrt{n}$ (assuming $m \leq n$). However, we found that for problems with large incoherence such a parameter setting *does not* provide exact recovery. Instead, we set $\lambda = \mu/\sqrt{n}$ in our experiments.

**Synthetic datasets:** Following the experimental setup of [CLMW11], the low-rank part $L^* = UV^T$ is generated using normally distributed $U \in \mathbb{R}^{m \times r}$, $V \in \mathbb{R}^{n \times r}$. Similarly, $supp(S^*)$ is generated by sampling a uniformly random subset of $[m] \times [n]$ with size $\|S^*\|_0$ and each non-zero $S_{ij}^*$ is drawn i.i.d. from the uniform distribution over $[r/(2\sqrt{mn}), r/\sqrt{mn}]$. For increasing incoherence of $L^*$, we randomly zero-out rows of $U, V$ and then re-normalize them.

There are three key problem parameters for RPCA with a fixed matrix size: a) sparsity of $S^*$, b) incoherence of $L^*$, c) rank of $L^*$. We investigate performance of both AltProj and IALM by varying each of the three parameters while fixing the others. In our plots (see Figure 2), we report computational time required by each of the two methods for decomposing $M$ into $L + S$ up to a relative error ($\|M - L - S\|_F / \|M\|_F$) of $10^{-3}$. Figure 2 shows that AltProj scales significantly better than IALM for increasingly dense $S^*$. We attribute this observation to the fact that as $\|S^*\|_0$ increases, the problem is "harder" and the intermediate iterates of IALM have ranks significantly larger than $r$. Our intuition is confirmed by Figure 2 (b), which shows that when density ($\alpha$) of $S^*$ is $0.4$ then the intermediate iterates of IALM can be of rank over $500$ while the rank of $L^*$ is only $5$. We observe a similar trend for the other parameters, i.e., AltProj scales significantly better than IALM with increasing incoherence parameter $\mu$ (Figure 2 (c)) and increasing rank (Figure 2 (d)). See Appendix C for additional plots.

**Real-world datasets:** Next, we apply our method to the problem of foreground-background (F-B) separation in a video [LHGT04]. The observed matrix $M$ is formed by vectorizing each frame and stacking them column-wise. Intuitively, the background in a video is the static part and hence forms a low-rank component while the foreground is a dynamic but sparse perturbation.

Here, we used two benchmark datasets named *Escalator* and *Restaurant* dataset. The *Escalator* dataset has $3417$ frames at a resolution of $160 \times 130$. We first applied the standard PCA method for extracting low-rank part. Figure 3 (b) shows the extracted background from the video. There are

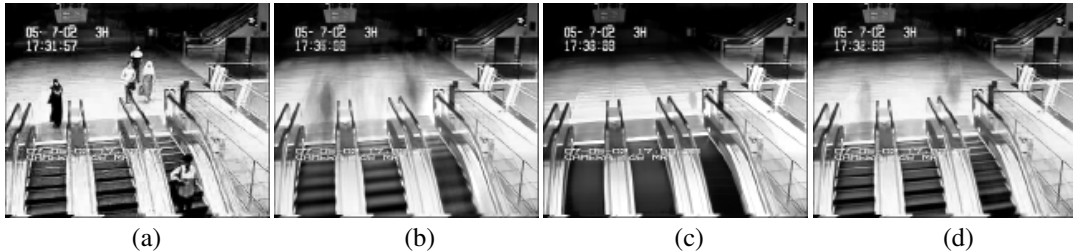

|       (a)       |       (b)       |       (c)       |       (d)       |

Figure 3: Foreground-background separation in the *Escalator* video. (a): Original image frame. (b): Best rank-10 approximation; time taken is $3.1s$. (c): Low-rank frame obtained using AltProj; time taken is $63.2s$. (d): Low-rank frame obtained using IALM; time taken is $1688.9s$.

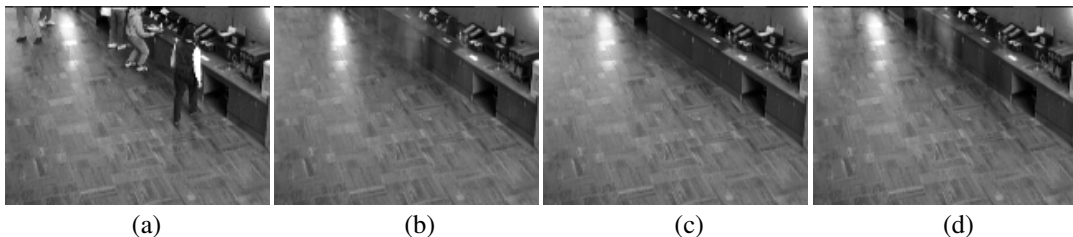

|       (a)       |       (b)       |       (c)       |       (d)       |

Figure 4: Foreground-background separation in the *Restaurant* video. (a): Original frame from the video. (b): Best rank-10 approximation (using PCA) of the original frame; $2.8s$ were required to compute the solution (c): Low-rank part obtained using AltProj; computational time required by AltProj was $34.9s$. (d): Low-rank part obtained using IALM; $693.2s$ required by IALM to compute the low-rank+sparse decomposition.

several artifacts (shadows of people near the escalator) that are not desirable. In contrast, both IALM and AltProj obtain significantly better F-B separation (see Figure 3(c), (d)). Interestingly, AltProj removes the steps of the escalator which are moving and arguably are part of the dynamic foreground, while IALM keeps the steps in the background part. Also, our method is significantly faster, i.e., our method, which takes $63.2s$ is about 26 times faster than IALM, which takes $1688.9s$.

*Restaurant dataset:* Figure 4 shows the comparison of AltProj and IALM on a subset of the "Restaurant" dataset where we consider the last 2055 frames at a resolution of $120 \times 160$. AltProj was around 19 times faster than IALM. Moreover, visually, the background extraction seems to be of better quality (for example, notice the blur near top corner counter in the IALM solution). Plot(b) shows the PCA solution and that also suffers from a similar blur at the top corner of the image, while the background frame extracted by AltProj does not have any noticeable artifacts.

## 5 Conclusion

In this work, we proposed a non-convex method for robust PCA, which consists of alternating projections on to low rank and sparse matrices. We established global convergence of our method under conditions which match those for convex methods. At the same time, our method has much faster running times, and has superior experimental performance. This work opens up a number of interesting questions for future investigation. While we match the convex methods, under the deterministic sparsity model, studying the random sparsity model is of interest. Our noisy recovery results assume deterministic noise; improving the results under random noise needs to be investigated. There are many decomposition problems beyond the robust PCA setting, e.g. structured sparsity models, robust tensor PCA problem, and so on. It is interesting to see if we can establish global convergence for non-convex methods in these settings.

## Acknowledgements

AA and UN would like to acknowledge NSF grant CCF-1219234, ONR N00014-14-1-0665, and Microsoft faculty fellowship. SS would like to acknowledge NSF grants 1302435, 0954059, 1017525 and DTRA grant HDTRA1-13-1-0024. PJ would like to acknowledge Nikhil Srivastava and Deeparnab Chakrabarty for several insightful discussions during the course of the project.

## Footnotes

*Part of the work done while interning at Microsoft Research, India

[1] If the input matrix $M$ is not symmetric, we embed it in a symmetric matrix and consider the eigenvectors of the corresponding matrix.

[2]$\epsilon$ is the desired reconstruction error

[3]Note that we only require a rank-$k$ approximation of the matrix rather than the actual singular vectors. Thus, the computational complexity has no dependence on the gap between the singular values.

[4]For instance, consider the $n \times n$ matrix which has $r$ copies of the all ones matrix, each of size $\frac{n}{r}$, placed across the diagonal. We see that this matrix has rank $r$ and is incoherent with parameter $\mu = 1$. Note that

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
