[Supplementary Material · rpca_supp.pdf]

# A Proof of Theorem 1

We will start with some preliminary lemmas. The first lemma is the well known Weyl's inequality in the matrix setting[Bha97].

**Lemma 2.** *Suppose $B = A + E$ be an $n \times n$ matrix. Let $\lambda_1, \cdots, \lambda_n$ and $\sigma_1, \cdots, \sigma_n$ be the eigenvalues of $B$ and $A$ respectively such that $\lambda_1 \geq \cdots \geq \lambda_n$ and $\sigma_1 \geq \cdots \geq \sigma_n$. Then we have:*

$$|\lambda_i - \sigma_i| \leq \|E\|_2 \ \forall \ i \in [n].$$

The following lemma is the Davis-Kahan theorem[Bha97], specialized for rank-1 matrices.

**Lemma 3.** *Suppose $B = A + E$. Let $A = \boldsymbol{u}^*(\boldsymbol{u}^*)^\top$ be a rank-1 matrix with unit spectral norm. Suppose further that $\|E\|_2 < \frac{1}{2}$. Then, we have:*

$$|\lambda - 1| < \|E\|_2, \ and$$
$$\left| \langle \boldsymbol{u}, \boldsymbol{u}^* \rangle^2 - 1 \right| < 4 \|E\|_2,$$

*where $\lambda$ and $\boldsymbol{u}$ are the top eigenvalue eigenvector pair of $B$.*

As outlined in Section 3.1 (and formalized in the proof of Theorem 1), it is sufficient to prove the correctness of Algorithm 1 for the case of symmetric matrices. So, most of the lemmas we prove in this section assume that the matrices are symmetric.

**Lemma 4.** *Let $S \in \mathbb{R}^{n \times n}$ satisfy assumption (S1). Then, $\|S\|_2 \leq \alpha n \|S\|_\infty$.*

*Proof of Lemma 4.* Let $x, y$ be unit vectors such that $\|S\|_2 = x^T S y = \sum_{ij} x_i y_j S_{ij}$. Then, using $a \cdot b \leq (a^2 + b^2)/2$, we have:

$$\|S\|_2 \leq \frac{1}{2} \sum_{ij} (x_i^2 + y_j^2) S_{ij} \leq \frac{1}{2}(\alpha n \|S\|_\infty + \alpha n \|S\|_\infty), \tag{3}$$

where the last inequality follows from the fact that $S$ has at most $\alpha n$ non-zeros per row and per column. □

**Lemma 5.** *Let $S \in \mathbb{R}^{n \times n}$ satisfy assumption (S1). Also, let $U \in \mathbb{R}^{n \times r}$ be a $\mu$-incoherent orthogonal matrix, i.e., $\max_i \left\| \boldsymbol{e}_i^\top U \right\|_2 \leq \frac{\mu \sqrt{r}}{\sqrt{n}}$, where $\boldsymbol{e}_i$ stands for the $i^{th}$ standard basis vector. Then, $\forall p \geq 0$, the following holds:*

$$\max_i \left\| \boldsymbol{e}_i^\top S^p U \right\|_2 \leq \frac{\mu \sqrt{r}}{\sqrt{n}} (\alpha \cdot n \cdot \|S\|_\infty)^p.$$

*Proof of Lemma 5.* We prove the lemma using mathematical induction.

Base Case ($p = 0$): This is just a restatement of the incoherence of $U$.

Induction step: We have:

$$\left\| \boldsymbol{e}_i^\top (S)^{p+1} U \right\|_2^2 = \|\boldsymbol{e}_i^\top S(S^p U)\|_2^2 = \sum_\ell (\boldsymbol{e}_i^\top S(S^p U) \boldsymbol{e}_\ell)^2 = \sum_\ell \Big( \sum_j S_{ij} \boldsymbol{e}_j^\top (S^p U) \boldsymbol{e}_\ell \Big)^2$$

$$= \sum_{j_1 j_2} S_{ij_1} S_{ij_2} \sum_\ell (\boldsymbol{e}_{j_1}^\top (S^p U) \boldsymbol{e}_\ell)(\boldsymbol{e}_\ell^\top (S^p U)^\top \boldsymbol{e}_{j_2})$$

$$\overset{\zeta_1}{\leq} \sum_{j_1 j_2} S_{ij_1} S_{ij_2} (\boldsymbol{e}_{j_1}^\top (S^p U)(S^p U)^\top \boldsymbol{e}_{j_2}) \leq \sum_{j_1 j_2} S_{ij_1} S_{ij_2} \|\boldsymbol{e}_{j_1}^T (S^p U)\|_2 \|\boldsymbol{e}_{j_2}^\top (S^p U)\|_2$$

$$\overset{\zeta_2}{\leq} \frac{\mu^2 r}{n} (\alpha \cdot n \cdot \|S\|_\infty)^{2p},$$

where $\zeta_1$ follows by $\sum_{\ell=1}^t \boldsymbol{e}_\ell \boldsymbol{e}_\ell^\top = I$, and $\zeta_2$ follows from assumption (S1) on $S$ and from the inductive hypothesis on $\left\| \boldsymbol{e}_i^\top S^p U \right\|_2$. □

In what follows, we prove a number of lemmas concerning the structure of $L^{(t)}$ and $E^{(t)} := S^* - S^{(t)}$. The following lemma shows that the threshold in (2) is close to that with $M - S^{(t)}$ replaced by $L^*$.

**Lemma 6.** *Let $L^*, S^*$ be symmetric and satisfy the assumptions of Theorem 1 and let $S^{(t)}$ be the $t^{th}$ iterate of the $k^{th}$ stage of Algorithm 1. Let $\sigma_1^*, \ldots, \sigma_r^*$ be the eigenvalues of $L^*$, such that $|\sigma_1^*| \geq \cdots \geq |\sigma_r^*|$ and $\lambda_1, \cdots, \lambda_n$ be the eigenvalues of $M - S^{(t)}$ such that $|\lambda_1| \geq \cdots \geq |\lambda_n|$. Recall that $E^{(t)} := S^* - S^{(t)}$. Suppose further that*

1. $\left\| E^{(t)} \right\|_\infty \leq \frac{8\mu^2 r}{n} \left( |\sigma_{k+1}^*| + \left(\frac{1}{2}\right)^{t-1} |\sigma_k^*| \right)$, *and*

2. $Supp\left(E^{(t)}\right) \subseteq Supp\left(S^*\right)$.

*Then,*

$$\frac{7}{8}\left( |\sigma_{k+1}^*| + \left(\frac{1}{2}\right)^t |\sigma_k^*| \right) \leq \left( |\lambda_{k+1}| + \left(\frac{1}{2}\right)^t |\lambda_k| \right) \leq \frac{9}{8}\left( |\sigma_{k+1}^*| + \left(\frac{1}{2}\right)^t |\sigma_k^*| \right). \quad (4)$$

*Proof.* Note that $M - S^{(t)} = L^* + E^{(t)}$. Now, using Lemmas 2 and 4, we have:

$$\left| \lambda_{k+1} - \sigma_{k+1}^* \right| \leq \left\| E^{(t)} \right\|_2 \leq \alpha n \left\| E^{(t)} \right\|_\infty \leq 8\mu^2 r \alpha \gamma_t,$$

where $\gamma_t := \left( |\sigma_{k+1}^*| + \left(\frac{1}{2}\right)^{t-1} |\sigma_k^*| \right)$. That is, $\left| |\lambda_{k+1}| - |\sigma_{k+1}^*| \right| \leq 8\mu^2 r \alpha \gamma_t$. Similarly, $\left| |\lambda_k| - |\sigma_k^*| \right| \leq 8\mu^2 r \alpha \gamma_t$. So we have:

$$\left| \left( |\lambda_{k+1}| + \left(\frac{1}{2}\right)^t |\lambda_k| \right) - \left( |\sigma_{k+1}^*| + \left(\frac{1}{2}\right)^t |\sigma_k^*| \right) \right| \leq 8\mu^2 r \alpha \gamma_t \left( 1 + \left(\frac{1}{2}\right)^t \right)$$

$$\leq 16\mu^2 r \alpha \gamma_t$$

$$\leq \frac{1}{8}\left( |\sigma_{k+1}^*| + \left(\frac{1}{2}\right)^t |\sigma_k^*| \right),$$

where the last inequality follows from the bound $\alpha \leq \frac{1}{512\mu^2 r}$. $\qquad \square$

The following lemma shows that under the same assumptions as in Lemma 6, we can obtain a bound on the $\ell_\infty$ norm of $L^{(t+1)} - L^*$. This is the most crucial step in our analysis since we bound $\ell_\infty$ norm of errors which are quite hard to obtain.

**Lemma 7.** *Assume the notation of Lemma 6. Also, let $L^{(t)}, S^{(t)}$ be the $t^{th}$ iterates of $k^{th}$ stage of Algorithm 1 and $L^{(t+1)}, S^{(t+1)}$ be the $(t+1)^{th}$ iterates of the same stage. Also, recall that $E^{(t)} := S^* - S^{(t)}$ and $E^{(t+1)} := S^* - S^{(t+1)}$. Suppose further that*

1. $\left\| E^{(t)} \right\|_\infty \leq \frac{8\mu^2 r}{n} \left( |\sigma_{k+1}^*| + \left(\frac{1}{2}\right)^{t-1} |\sigma_k^*| \right)$, *and*

2. $Supp\left(E^{(t)}\right) \subseteq Supp\left(S^*\right)$.

*Then, we have:*

$$\left\| L^{(t+1)} - L^* \right\|_\infty \leq \frac{2\mu^2 r}{n}\left( |\sigma_{k+1}^*| + \left(\frac{1}{2}\right)^t |\sigma_k^*| \right).$$

*Proof.* Let $L^{(t+1)} = P_k(M - S^{(t)}) = U\Lambda U^\top$ be the eigenvalue decomposition of $L^{(t+1)}$. Also, recall that $M - S^{(t)} = L^* + E^{(t)}$. Then, for every eigenvector $\boldsymbol{u}_i$ of $L^{(t+1)}$, we have

$$\left(L^* + E^{(t)}\right)\boldsymbol{u}_i = \lambda_i \boldsymbol{u}_i,$$

$$\left(I - \frac{E^{(t)}}{\lambda_i}\right)\boldsymbol{u}_i = \frac{1}{\lambda_i}L^*\boldsymbol{u}_i,$$

$$\boldsymbol{u}_i = \left(I - \frac{E^{(t)}}{\lambda_i}\right)^{-1}\frac{L^*\boldsymbol{u}_i}{\lambda_i}$$

$$= \left(I + \frac{E^{(t)}}{\lambda_i} + \left(\frac{E^{(t)}}{\lambda_i}\right)^2 + \dots\right)\frac{L^*\boldsymbol{u}_i}{\lambda_i}. \tag{5}$$

Note that we used Lemmas 2 and 4 to guarantee the existence of $\left(I - \frac{E^{(t)}}{\lambda_i}\right)^{-1}$. Hence,

$$U\Lambda U^\top - L^* = \left(L^*U\Lambda^{-1}U^\top L^* - L^*\right) + \sum_{p+q\geq 1}\left(E^{(t)}\right)^p L^*U\Lambda^{-(p+q+1)}U^\top L^*\left(E^{(t)}\right)^q.$$

By triangle inequality, we have

$$\left\|U\Lambda U^\top - L^*\right\|_\infty \leq \left\|L^*U\Lambda^{-1}U^\top L^* - L^*\right\|_\infty$$
$$+ \sum_{p+q\geq 1}\left\|\left(E^{(t)}\right)^p L^*U\Lambda^{-(p+q+1)}U^\top L^*\left(E^{(t)}\right)^q\right\|_\infty. \tag{6}$$

We now bound the two terms on the right hand side above.

We note that,

$$\left\|L^*U\Lambda^{-1}U^\top L^* - L^*\right\|_\infty$$
$$= \max_{ij}\boldsymbol{e}_i^\top\left(U^*\Sigma^*(U^*)^\top U\Lambda^{-1}U^\top U^*\Sigma^*(U^*)^\top - U^*\Sigma^*(U^*)^\top\right)\boldsymbol{e}_j$$
$$= \max_{ij}\boldsymbol{e}_i^\top U^*\left(\Sigma^*(U^*)^\top U\Lambda^{-1}U^\top U^*\Sigma^* - \Sigma^*\right)(U^*)^\top\boldsymbol{e}_j$$
$$\leq \max_{ij}\left\|\boldsymbol{e}_i^\top U^*\right\|\cdot\left\|\boldsymbol{e}_j^\top U^*\right\|\cdot\left\|U^*\Sigma^*(U^*)^\top U\Lambda^{-1}U^\top U^*\Sigma^*(U^*)^\top - U^*\Sigma^*(U^*)^\top\right\|_2$$
$$\leq \frac{\mu^2 r}{n}\left\|L^*U\Lambda^{-1}U^\top L^* - L^*\right\|_2, \tag{7}$$

where we denote $U^*\Sigma^*(U^*)^\top$ to be the SVD of $L^*$. Let $L^* + E^{(t)} = U\Lambda U^\top + \widetilde{U}\widetilde{\Lambda}\widetilde{U}^\top$ be the eigenvalue decomposition of $L^* + E^{(t)}$. Note that $\widetilde{U}^\top U = 0$. Recall that, $U\Lambda U^\top = P_k(M - S^{(t)}) = P_k(L^* + E^{(t)}) = L^{(t+1)}$. Also note that,

$$L^*U\Lambda^{-1}U^\top L^* - L^*$$
$$= \left(U\Lambda U^\top + \widetilde{U}\widetilde{\Lambda}\widetilde{U}^\top - E^{(t)}\right)U\Lambda^{-1}U^\top\left(U\Lambda U^\top + \widetilde{U}\widetilde{\Lambda}\widetilde{U}^T - E^{(t)}\right) - L^*,$$
$$= \left(UU^\top - \left(E^{(t)}\right)U\Lambda^{-1}U^\top\right)\left(U\Lambda U^\top + \widetilde{U}\widetilde{\Lambda}\widetilde{U}^T - E^{(t)}\right) - L^*,$$
$$= -UU^\top E^{(t)} - E^{(t)}UU^\top - E^{(t)}U\Lambda^{-1}U^\top E^{(t)^\top} - \widetilde{U}\widetilde{\Lambda}\widetilde{U}^\top + E^{(t)}. \tag{8}$$

Hence, using Lemma 8, we have:

$$\left\|L^*U\Lambda^{-1}U^\top L^* - L^*\right\|_2 \leq 3\left\|E^{(t)}\right\|_2 + \frac{\left\|E^{(t)}\right\|_2^2}{|\lambda_k|} + |\lambda_{k+1}|$$
$$\leq |\sigma^*_{k+1}| + 5\left\|E^{(t)}\right\|_2. \tag{9}$$

Combining (7) and (9), we have:

$$\left\|L^*U\Lambda^{-1}U^\top L^* - L^*\right\|_\infty \leq \frac{\mu^2 r}{n}\left(|\sigma^*_{k+1}| + 5\left\|E^{(t)}\right\|_2\right) \tag{10}$$

Now, we will bound the $(p, q)^{\text{th}}$ term of $\sum_{p+q \geq 1} \left\| \left(E^{(t)}\right)^p L^* U \Lambda^{-(p+q+1)} U^\top L^* \left(E^{(t)}\right)^q \right\|_\infty$:

$$\left\| (E^{(t)})^p L^* U \Lambda^{-(p+q+1)} U^\top L^* (E^{(t)})^q \right\|_\infty$$

$$= \max_{ij} \boldsymbol{e}_i^\top \left( (E^{(t)})^p L^* U \Lambda^{-(p+q+1)} U^\top L^* (E^{(t)})^q \right) \boldsymbol{e}_j,$$

$$\leq \max_{ij} \left\| \boldsymbol{e}_i^\top (E^{(t)})^p U^* \right\|_2 \left\| \boldsymbol{e}_j^\top (E^{(t)})^q U^* \right\|_2 \left\| L^* U \Lambda^{-(p+q+1)} U^\top L^* \right\|_2,$$

$$\overset{\zeta_1}{\leq} \frac{\mu^2 r}{n} \left( \alpha n \left\| E^{(t)} \right\|_\infty \right)^p \left( \alpha n \left\| E^{(t)} \right\|_\infty \right)^q \left\| L^* U \Lambda^{-(p+q+1)} U^\top L^* \right\|_2, \qquad (11)$$

where $\zeta_1$ follows from Lemma 5 and the incoherence of $L^*$. Now, similar to (8), we have:

$$\left\| L^* U \Lambda^{-(p+q+1)} U^\top L^* \right\|_2$$

$$= \left\| U \Lambda^{-(p+q-1)} U^\top - E^{(t)} U \Lambda^{-(p+q)} U^\top - U \Lambda^{-(p+q)} U^\top E^{(t)} + E^{(t)} U \Lambda^{-(p+q+1)} U^\top E^{(t)} \right\|_2,$$

$$\leq \| \Lambda^{-(p+q-1)} \|_2 + 2 \| E^{(t)} \|_2 \| \Lambda^{-(p+q)} \|_2 + \| E^{(t)} \|_2^2 \| \Lambda^{-(p+q+1)} \|_2,$$

$$\leq |\lambda_k|^{-(p+q-1)} \left( 1 + 2 \frac{\| E^{(t)} \|_2}{|\lambda_k|} + \frac{\| E^{(t)} \|_2^2}{\lambda_k^2} \right) = |\lambda_k|^{-(p+q-1)} \left( 1 + \frac{\| E^{(t)} \|_2}{|\lambda_k|} \right)^2,$$

$$\leq |\lambda_k|^{-(p+q-1)} \left( 1 + \frac{\| E^{(t)} \|_2}{|\lambda_k|} \right)^2,$$

$$\overset{\zeta_1}{\leq} |\lambda_k|^{-(p+q-1)} \left( 1 + \frac{17 \mu^2 r \alpha |\sigma_k^*|}{(1 - 17 \mu^2 r \alpha) |\sigma_k^*|} \right)^2 \leq 2 |\lambda_k|^{-(p+q-1)}, \qquad (12)$$

where $\zeta_1$ follows from Lemma 8.

Using (11), (12), we have:

$$\left\| (E^{(t)})^p L^* U \Lambda^{-(p+q+1)} U^\top L^* (E^{(t)})^q \right\|_\infty \leq 2 \alpha \mu^2 r \left\| E^{(t)} \right\|_\infty \left( \frac{\alpha n \left\| E^{(t)} \right\|_\infty}{|\lambda_k|} \right)^{p+q-1}. \qquad (13)$$

Using the above bound, and the assumption on $\left\| E^{(t)} \right\|_\infty$:

$$\left\| E^{(t)} \right\|_\infty \leq \frac{8 \mu^2 r}{n} \left( |\sigma_{k+1}^*| + \left( \frac{1}{2} \right)^{t-1} |\sigma_k^*| \right) \leq \frac{17 \mu^2 r}{n} |\sigma_k^*|,$$

we have:

$$\sum_{p+q \geq 1} \left\| \left( E^{(t)} \right)^p L^* U \Lambda^{-(p+q+1)} U^\top L^* \left( E^{(t)} \right)^q \right\|_\infty$$

$$\leq 2 \mu^2 r \alpha \left\| E^{(t)} \right\|_\infty \sum_{p+q \geq 1} \left( \frac{\alpha n \left\| E^{(t)} \right\|_\infty}{|\lambda_k|} \right)^{p+q-1}$$

$$\leq 2 \mu^2 r \alpha \left\| E^{(t)} \right\|_\infty \left( \frac{1}{1 - \frac{17 \mu^2 \alpha r}{1 - 17 \mu^2 \alpha \cdot r}} \right)^2$$

$$\leq 2 \mu^2 r \alpha \left\| E^{(t)} \right\|_\infty \left( \frac{1}{1 - 34 \mu^2 r \alpha} \right)^2$$

$$\leq 4 \mu^2 r \alpha \left\| E^{(t)} \right\|_\infty. \qquad (14)$$

Combining (6), (10), (14), we have:

$$\| U \Lambda U^\top - L^* \|_\infty \leq \frac{\mu^2 r}{n} \left( |\sigma_{k+1}^*| + 5 \left\| E^{(t)} \right\|_2 + 4 \mu^2 r \alpha n \left\| E^{(t)} \right\|_\infty \right)$$

$$\leq \frac{2 \mu^2 r}{n} \left( |\sigma_{k+1}^*| + \left( \frac{1}{2} \right)^t |\sigma_k^*| \right),$$

where we used Lemma 4 and the assumption on $\left\|E^{(t)}\right\|_\infty$. □

We used the following technical lemma in the proof of Lemma 7.

**Lemma 8.** *Assume the notation of Lemma 7. Suppose further that*

1. $\left\|E^{(t)}\right\|_\infty \leq \frac{8\mu^2 r}{n}\left(|\sigma_{k+1}^*| + \left(\frac{1}{2}\right)^{t-1}|\sigma_k^*|\right)$, *and*

2. $Supp\left(E^{(t)}\right) \subseteq Supp\left(S^*\right)$.

*Then we have:*

$$\left\|E^{(t)}\right\|_2 \leq 17\mu^2 r\alpha|\sigma_k^*|, \quad |\lambda_k| \geq |\sigma_k^*|\left(1 - 17\mu^2 r\alpha\right), \ \ and \ \ |\lambda_{k+1}| \leq |\sigma_{k+1}^*| + \left\|E^{(t)}\right\|_2.$$

*Proof.* Using Lemmas 4 and 2, we have:

$$|\lambda_i - \sigma_i^*| \leq \|E^{(t)}\|_2 \leq \alpha n\left\|E^{(t)}\right\|_\infty.$$

The result follows by using the bound on $\left\|E^{(t)}\right\|_\infty$. □

The following lemma bounds the support of $E^{(t+1)}$ and $\left\|E^{(t+1)}\right\|_\infty$, using an assumption on $\left\|L^{(t+1)} - L^*\right\|_\infty$.

**Lemma 9.** *Assume the notation of Lemma 7. Suppose*

$$\left\|L^{(t+1)} - L^*\right\|_\infty \leq \frac{2\mu^2 r}{n}\left(|\sigma_{k+1}^*| + \left(\frac{1}{2}\right)^t|\sigma_k^*|\right).$$

*Then, we have:*

1. $Supp\left(E^{(t+1)}\right) \subseteq Supp\left(S^*\right)$.

2. $\left\|E^{(t+1)}\right\|_\infty \leq \frac{7\mu^2 r}{n}\left(|\sigma_{k+1}^*| + \left(\frac{1}{2}\right)^t|\sigma_k^*|\right)$, *and*

*Proof.* We first prove the first conclusion. Recall that,

$$S^{(t+1)} = H_\zeta(M - L^{(t+1)}) = H_\zeta(L^* - L^{(t+1)} + S^*),$$

where $\zeta = \frac{4\mu^2 r}{n}\left(|\lambda_{k+1}| + \left(\frac{1}{2}\right)^t|\lambda_k|\right)$ is as defined in Algorithm 1 and $\lambda_1, \cdots, \lambda_n$ are the eigenvalues of $M - S^{(t)}$ such that $|\lambda_1| \geq \cdots \geq |\lambda_n|$.

If $S_{ij}^* = 0$ then $E_{ij}^{(t+1)} = \mathbb{1}_{\left\{\left|L_{ij}^* - L_{ij}^{(t+1)}\right| > \zeta\right\}} \cdot (L_{ij}^* - L_{ij}^{(t+1)})$. The first part of the lemma now follows by using the assumption that $\left\|L^{(t+1)} - L^*\right\|_\infty \leq \frac{2\mu^2 r}{n}\left(|\sigma_{k+1}^*| + \left(\frac{1}{2}\right)^t|\sigma_k^*|\right) \overset{(\zeta_1)}{\leq} \frac{4\mu^2 r}{n}\left(|\lambda_{k+1}^*| + \left(\frac{1}{2}\right)^t|\lambda_k^*|\right) = \zeta$, where $(\zeta_1)$ follows from Lemma 6.

We now prove the second conclusion. We consider the following two cases:

1. $\left|M_{ij} - L_{ij}^{(t+1)}\right| > \zeta$: Here, $S_{ij}^{(t+1)} = S_{ij}^* + L_{ij}^* - L_{ij}^{(t+1)}$. Hence, $|S_{ij}^{(t+1)} - S_{ij}^*| \leq |L_{ij}^* - L_{ij}^{(t+1)}| \leq \frac{2\mu^2 r}{n}\left(|\sigma_{k+1}^*| + \left(\frac{1}{2}\right)^t|\sigma_k^*|\right)$.

2. $\left|M_{ij} - L_{ij}^{(t+1)}\right| \leq \zeta$: In this case, $S_{ij}^{(t+1)} = 0$ and $\left|S_{ij}^* + L_{ij}^* - L_{ij}^{(t+1)}\right| \leq \zeta$. So we have, $\left|E_{ij}^{(t+1)}\right| = |S_{ij}^*| \leq \zeta + \left|L_{ij}^* - L_{ij}^{(t+1)}\right| \leq \frac{7\mu^2 r}{n}\left(|\sigma_{k+1}^*| + \left(\frac{1}{2}\right)^t|\sigma_k^*|\right)$. The last inequality above follows from Lemma 6.

This proves the lemma. □

We are now ready to prove Lemma 1. In fact, we prove the following stronger version.

*Proof of Lemma 1.* Recall that in the $k^{\text{th}}$ stage, the update $L^{(t+1)}$ is given by: $L^{(t+1)} = P_k(M - S^{(t)})$ and $S^{(t+1)}$ is given by: $S^{(t+1)} = H_\zeta(M - L^{(t+1)})$. Also, recall that $E^{(t)} := S^* - S^{(t)}$ and $E^{(t+1)} := S^* - S^{(t+1)}$.

We prove the lemma by induction on both $k$ and $t$. For the base case ($k = 1$ and $t = -1$), we first note that the first inequality on $\left\|L^{(0)} - L^*\right\|_\infty$ is trivially satisfied. Due to the thresholding step (step 3 in Algorithm 1) and the incoherence assumption on $L^*$, we have:

$$\left\|E^{(0)}\right\|_\infty \leq \frac{8\mu^2 r}{n} \left(\sigma_2^* + 2\sigma_1^*\right), \text{ and}$$
$$\text{Supp}\left(E^{(0)}\right) \subseteq \text{Supp}\left(S^*\right).$$

So the base case of induction is satisfied.

We first do the inductive step over $t$ (for a fixed $k$). By inductive hypothesis we assume that: a) $\left\|E^{(t)}\right\|_\infty \leq \frac{8\mu^2 r}{n}\left(|\sigma_{k+1}^*| + \left(\frac{1}{2}\right)^{t-1}|\sigma_k^*|\right)$, b) $\text{Supp}\left(E^{(t)}\right) \subseteq \text{Supp}\left(S^*\right)$. Then by Lemma 7, we have:

$$\left\|L^{(t+1)} - L^*\right\|_\infty \leq \frac{2\mu^2 r}{n}\left(|\sigma_{k+1}^*| + \left(\frac{1}{2}\right)^{t+1}|\sigma_k^*|\right).$$

Lemma 9 now tells us that

1. $\left\|E^{(t+1)}\right\|_\infty \leq \frac{8\mu^2 r}{n}\left(|\sigma_{k+1}^*| + \left(\frac{1}{2}\right)^t |\sigma_k^*|\right)$, and

2. $\text{Supp}\left(E^{(t+1)}\right) \subseteq \text{Supp}\left(S^*\right)$.

This finishes the induction over $t$. Note that we show a stronger bound than necessary on $\left\|E^{(t+1)}\right\|_\infty$.

We now do the induction over $k$. Suppose the hypothesis holds for stage $k$. Let $T$ denote the number of iterations in each stage. We first obtain a lower bound on $T$. Since

$$\left\|M - S^{(0)}\right\|_2 \geq \|L^*\|_2 - \left\|E^{(0)}\right\|_2 \geq |\sigma_1^*| - \alpha n \left\|E^{(0)}\right\|_\infty \geq \frac{3}{4}|\sigma_1^*|,$$

we see that $T \geq 10 \log\left(3\mu^2 r |\sigma_1^*| / \epsilon\right)$. So, at the end of stage $k$, we have:

1. $\left\|E^{(T)}\right\|_\infty \leq \frac{7\mu^2 r}{n}\left(|\sigma_{k+1}^*| + \left(\frac{1}{2}\right)^T |\sigma_k^*|\right) \leq \frac{7\mu^2 r |\sigma_{k+1}^*|}{n} + \frac{\epsilon}{10n}$, and

2. $\text{Supp}\left(E^{(T)}\right) \subseteq \text{Supp}\left(S^*\right)$.

Lemmas 4 and 2 tell us that $\left|\sigma_{k+1}\left(M - S^{(T)}\right) - |\sigma_{k+1}^*|\right| \leq \left\|E^{(T)}\right\|_2 \leq \alpha\left(7\mu^2 r |\sigma_{k+1}^*| + \epsilon\right)$. We will now consider two cases:

1. **Algorithm 1 terminates:** This means that $\beta\sigma_{k+1}\left(M - S^{(T)}\right) < \frac{\epsilon}{2n}$ which then implies that $|\sigma_{k+1}^*| < \frac{\epsilon}{6\mu^2 r}$. So we have:

$$\left\|\widehat{L} - L^*\right\|_\infty = \left\|L^{(T)} - L^*\right\|_\infty \leq \frac{2\mu^2 r}{n}\left(|\sigma_{k+1}^*| + \left(\frac{1}{2}\right)^T |\sigma_k^*|\right) \leq \frac{\epsilon}{5n}.$$

This proves the statement about $\widehat{L}$. A similar argument proves the claim on $\left\|\widehat{S} - S^*\right\|_\infty$. The claim on $\text{Supp}\left(\widehat{S}\right)$ follows since $\text{Supp}\left(E^{(T)}\right) \subseteq \text{Supp}\left(S^*\right)$.

2. **Algorithm 1 continues to stage $(k+1)$:** This means that $\beta \sigma_{k+1}\left(L^{(T)}\right) \geq \frac{\epsilon}{2n}$ which then implies that $\left|\sigma_{k+1}^*\right| > \frac{\epsilon}{8\mu^2 r}$. So we have:

$$\left\|E^{(T)}\right\|_\infty \leq \frac{8\mu^2 r}{n}\left(\left|\sigma_{k+1}^*\right| + \left(\frac{1}{2}\right)^T \left|\sigma_k^*\right|\right)$$

$$\leq \frac{8\mu^2 r}{n}\left(\left|\sigma_{k+1}^*\right| + \frac{\epsilon}{10\mu^2 rn}\right)$$

$$\leq \frac{8\mu^2 r}{n}\left(\left|\sigma_{k+1}^*\right| + \frac{8\left|\sigma_{k+1}^*\right|}{10n}\right)$$

$$\leq \frac{8\mu^2 r}{n}\left(\left|\sigma_{k+2}^*\right| + 2\left|\sigma_{k+1}^*\right|\right).$$

Similarly for $\left\|L^{(T)} - L^*\right\|_\infty$.

This finishes the proof. □

*Proof of Theorem 1.* Using Lemma 1, it suffices to show that the general case can be reduced to the case of symmetric matrices. We will now outline this reduction.

Recall that we are given an $m \times n$ matrix $M = L^* + S^*$ where $L^*$ is the true low-rank matrix and $S^*$ the sparse error matrix. Wlog, let $m \leq n$ and suppose $\beta m \leq n < (\beta + 1)m$, for some $\beta \geq 1$. We then consider the symmetric matrices

$$\widetilde{M} = \begin{bmatrix} 0 & 0 & M \\ \vdots & \cdots & \vdots & \vdots \\ 0 & 0 & M \\ M^\top & \cdots & M^\top & 0 \end{bmatrix}, \widetilde{L} = \begin{bmatrix} 0 & 0 & L^* \\ \vdots & \cdots & \vdots & \vdots \\ 0 & 0 & L^* \\ (L^*)^\top & \cdots & (L^*)^\top & 0 \end{bmatrix}, \quad (15)$$

$$\underbrace{\phantom{xxxxxxxxxxx}}_{\beta \text{ times}} \qquad \underbrace{\phantom{xxxxxxxxxxx}}_{\beta \text{ times}}$$

and $\widetilde{S} = \widetilde{M} - \widetilde{L}$. A simple calculation shows that $\widetilde{L}$ is incoherent with parameter $\sqrt{3}\mu$ and $\widetilde{S}$ satisfies the sparsity condition (S1) with parameter $\frac{\alpha}{\sqrt{2}}$. Moreover the iterates of AltProj with input $\widetilde{M}$ have similar expressions as in (15) in terms of the corresponding iterates with input $M$. This means that it suffices to obtain the same guarantees for Algorithm 1 for the symmetric case. Lemma 1 does precisely this, proving the theorem. □

## B  Proof of Theorem 2

In this section, we prove Theorem 2. The roadmap of the proofs in this section is essentially the same as that in Appendix A.

In what follows, we prove a number of lemmas concerning the structure of $L^{(t)}$ and $E^{(t)} := S^* - S^{(t)}$. The first lemma is a generalization of Lemma 6 and shows that the threshold in (2) is close to that with $M^{(t)}$ replaced by $L^*$.

**Lemma 10.** *Let $L^*, S^*, N^*$ be symmetric and satisfy the assumptions of Theorem 2 and let $S^{(t)}$ be the $t^{th}$ iterate of the $k^{th}$ stage of Algorithm 1. Let $\sigma_1^*, \ldots, \sigma_r^*$ be the eigenvalues of $L^*$, such that $|\sigma_1^*| \geq \cdots \geq |\sigma_r^*|$ and $\lambda_1, \cdots, \lambda_n$ be the eigenvalues of $M - S^{(t)}$ such that $|\lambda_1| \geq \cdots \geq |\lambda_n|$. Recall that $E^{(t)} := S^* - S^{(t)}$. Suppose further that*

1. $\left\|E^{(t)}\right\|_\infty \leq \frac{8\mu^2 r}{n}\left(\left|\sigma_{k+1}^*\right| + \left(\frac{1}{2}\right)^{t-1}\left|\sigma_k^*\right| + 7\left\|N^*\right\|_2 + \frac{8n}{\sqrt{r}}\left\|N^*\right\|_\infty\right)$, *and*

2. *Supp $\left(S^{(t)}\right) \subseteq$ Supp $(S^*)$.*

*Then,*

$$\frac{7}{8}\left(\left|\sigma_{k+1}^*\right| + \left(\frac{1}{2}\right)^t\left|\sigma_k^*\right|\right) \leq \left(\left|\lambda_{k+1}\right| + \left(\frac{1}{2}\right)^t\left|\lambda_k\right|\right) \leq \frac{9}{8}\left(\left|\sigma_{k+1}^*\right| + \left(\frac{1}{2}\right)^t\left|\sigma_k^*\right|\right). \quad (16)$$

*Proof.* Note that $M - S^{(t)} = L^* + N^* + E^{(t)}$. Now, using Lemmas 2 and 4, we have:

$$\left| \lambda_{k+1} - \sigma_{k+1}^* \right| \leq \left\| E^{(t)} \right\|_2 \leq \alpha n \left\| E^{(t)} \right\|_\infty \leq 8\mu^2 r \alpha \gamma_t,$$

where $\gamma_t = \left( |\sigma_{k+1}^*| + \left(\frac{1}{2}\right)^{t-1} |\sigma_k^*| + 7 \|N^*\|_2 + \frac{8n}{\sqrt{r}} \|N^*\|_\infty \right)$. That is, $\left| |\lambda_{k+1}| - |\sigma_{k+1}^*| \right| \leq 8\mu^2 r \alpha \gamma_t$. Similarly, $||\lambda_k| - |\sigma_k^*|| \leq 8\mu^2 r \alpha \gamma_t$. So we have:

$$\left| \left( |\lambda_{k+1}| + \left(\frac{1}{2}\right)^t |\lambda_k| \right) - \left( |\sigma_{k+1}^*| + \left(\frac{1}{2}\right)^t |\sigma_k^*| \right) \right| \leq 8\mu^2 r \alpha \gamma_t \left( 1 + \left(\frac{1}{2}\right)^t \right)$$

$$\leq 16\mu^2 r \alpha \gamma_t$$

$$\leq \frac{1}{8} \left( |\sigma_{k+1}^*| + \left(\frac{1}{2}\right)^t |\sigma_k^*| \right),$$

where the last inequality follows from the bound $\alpha \leq \frac{1}{512\mu^2 r}$ and the assumption on $\|N^*\|_\infty$. $\qquad\square$

The following lemma shows that under the same assumptions as in Lemma 6, we can obtain a bound on the $\ell_\infty$ norm of $L^{(t+1)} - L^*$. This is the most crucial step in our analysis since we bound $\ell_\infty$ norm of errors which are quite hard to obtain.

**Lemma 11.** *Assume the notation of Lemma 6. Also, let $L^{(t)}, S^{(t)}$ be the $t^{th}$ iterates of $k^{th}$ stage of Algorithm 1 and $L^{(t+1)}, S^{(t+1)}$ be the $(t+1)^{th}$ iterates of the same stage. Also, recall that $E^{(t)} := S^* - S^{(t)}$ and $E^{(t+1)} := S^* - S^{(t+1)}$. Suppose further that*

1. $\left\| E^{(t)} \right\|_\infty \leq \frac{8\mu^2 r}{n} \left( |\sigma_{k+1}^*| + \left(\frac{1}{2}\right)^{t-1} |\sigma_k^*| + 7 \|N^*\|_2 + \frac{8n}{\sqrt{r}} \|N^*\|_\infty \right)$, *and*

2. $Supp\left( E^{(t)} \right) \subseteq Supp\left( S^* \right)$.

*Then, we have:*

$$\left\| L^{(t+1)} - L^* \right\|_\infty \leq \frac{2\mu^2 r}{n} \left( |\sigma_{k+1}^*| + \left(\frac{1}{2}\right)^t |\sigma_k^*| + 7 \|N^*\|_2 + \frac{8n}{\sqrt{r}} \|N^*\|_\infty \right).$$

*Proof.* Let $L^{(t+1)} = P_k(M - S^{(t)}) = U \Lambda U^\top$ be the eigenvalue decomposition of $L^{(t+1)}$. Also, recall that $M - S^{(t)} = L^* + N^* + E^{(t)}$. Then, for every eigenvector $\boldsymbol{u}_i$ of $L^{(t+1)}$, we have

$$\left( L^* + N^* + E^{(t)} \right) \boldsymbol{u}_i = \lambda_i \boldsymbol{u}_i,$$

$$\left( I - \frac{E^{(t)}}{\lambda_i} \right) \boldsymbol{u}_i = \frac{1}{\lambda_i} \left( L^* + N^* \right) \boldsymbol{u}_i,$$

$$\boldsymbol{u}_i = \left( I - \frac{E^{(t)}}{\lambda_i} \right)^{-1} \frac{\left( L^* + N^* \right) \boldsymbol{u}_i}{\lambda_i}$$

$$= \left( I + \frac{E^{(t)}}{\lambda_i} + \left( \frac{E^{(t)}}{\lambda_i} \right)^2 + \dots \right) \frac{\left( L^* + N^* \right) \boldsymbol{u}_i}{\lambda_i}.$$

Note that we used Lemmas 2 and 4 to guarantee the existence of $\left( I - \frac{E^{(t)}}{\lambda_i} \right)^{-1}$. Hence,

$$U \Lambda U^\top - L^* = \left( \left( L^* + N^* \right) U \Lambda^{-1} U^\top \left( L^* + N^* \right) - L^* \right)$$

$$+ \sum_{p+q \geq 1} (S^{(t)})^p \left( L^* + N^* \right) U \Lambda^{-(p+q+1)} U^\top \left( L^* + N^* \right) (S^{(t)})^q.$$

By triangle inequality, we have

$$\left\| U \Lambda U^\top - L^* \right\|_\infty \leq \left\| \left( L^* + N^* \right) U \Lambda^{-1} U^\top \left( L^* + N^* \right) - L^* \right\|_\infty$$

$$+ \sum_{p+q \geq 1} \left\| (S^{(t)})^p \left( L^* + N^* \right) U \Lambda^{-(p+q+1)} U^\top \left( L^* + N^* \right) (S^{(t)})^q \right\|_\infty. \quad (17)$$

We now bound the two terms on the right hand side above.

For the first term, we again use triangle inequality to obtain

$$\left\| (L^* + N^*) U\Lambda^{-1} U^\top (L^* + N^*) - L^* \right\|_\infty \leq \left\| L^* U\Lambda^{-1} U^\top L^* - L^* \right\|_\infty + \left\| N^* U\Lambda^{-1} U^\top L^* \right\|_\infty$$
$$+ \left\| L^* U\Lambda^{-1} U^\top N^* \right\|_\infty + \left\| N^* U\Lambda^{-1} U^\top N^* \right\|_\infty .$$
(18)

We note that,

$$\left\| L^* U\Lambda^{-1} U^\top L^* - L^* \right\|_\infty$$
$$= \max_{ij} \boldsymbol{e}_i{}^\top \left( U^*\Sigma^*(U^*)^\top U\Lambda^{-1} U^\top U^*\Sigma^*(U^*)^\top - U^*\Sigma^*(U^*)^\top \right) \boldsymbol{e}_j$$
$$= \max_{ij} \boldsymbol{e}_i{}^\top U^* \left( \Sigma^*(U^*)^\top U\Lambda^{-1} U^\top U^*\Sigma^* - \Sigma^* \right)(U^*)^\top \boldsymbol{e}_j$$
$$\leq \max_{ij} \|\boldsymbol{e}_i{}^\top U^*\| \cdot \|\boldsymbol{e}_j{}^\top U^*\| \cdot \|U^*\Sigma^*(U^*)^\top U\Lambda^{-1} U^\top U^*\Sigma^*(U^*)^\top - U^*\Sigma^*(U^*)^\top\|_2$$
$$\leq \frac{\mu^2 r}{n} \|L^* U\Lambda^{-1} U^\top L^* - L^*\|_2,$$
(19)

where we denote $U^*\Sigma^*(U^*)^\top$ to be the SVD of $L^*$. Let $L^* + N^* + E^{(t)} = U\Lambda U^\top + \widetilde{U}\widetilde{\Lambda}\widetilde{U}^\top$ be the eigenvalue decomposition of $L^* + N^* + E^{(t)}$. Note that $\widetilde{U}^\top U = 0$. Recall that, $U\Lambda U^\top = P_k(M^{(t)}) = L^{(t)}$. Also note that,

$$L^* U\Lambda^{-1} U^\top L^* - L^*$$
$$= (U\Lambda U^\top + \widetilde{U}\widetilde{\Lambda}\widetilde{U}^\top - N^* - E^{(t)})U\Lambda^{-1} U^\top (U\Lambda U^\top + \widetilde{U}\widetilde{\Lambda}\widetilde{U}^\top - N^* - E^{(t)}) - L^*,$$
$$= (UU^\top - \left( N^* + E^{(t)} \right) U\Lambda^{-1} U^\top)(U\Lambda U^\top + \widetilde{U}\widetilde{\Lambda}\widetilde{U}^\top - N^* - E^{(t)}) - L^*,$$
$$= U\Lambda U^\top - UU^\top \left( N^* + E^{(t)} \right) - \left( N^* + E^{(t)} \right) UU^\top$$
$$- \left( N^* + E^{(t)} \right) U\Lambda^{-1} U^\top \left( N^* + E^{(t)} \right)^\top - U\Lambda U^\top - \widetilde{U}\widetilde{\Lambda}\widetilde{U}^\top + N^* + E^{(t)}.$$
(20)

Hence, using Lemma 12, we have:

$$\left\| L^* U\Lambda^{-1} U^\top L^* - L^* \right\|_2 \leq 3 \left\| N^* + E^{(t)} \right\|_2 + \frac{\left\| N^* + E^{(t)} \right\|_2^2}{|\lambda_k|} + |\lambda_{k+1}|$$

$$\leq \left| \sigma_{k+1}^* \right| + 4 \left\| N^* + E^{(t)} \right\|_2 + \frac{\left\| N^* + E^{(t)} \right\|_2^2}{(1 - 17\mu^2 r\alpha) \left| \sigma_k^* \right|}.$$
(21)

Using (19), (21), and Lemma 12:

$$\left\| L^* U\Lambda^{-1} U^\top L^* - L^* \right\|_\infty \leq \frac{\mu^2 r}{n} \left( \left| \sigma_{k+1}^* \right| + 7 \|N^*\|_2 + 5 \left\| E^{(t)} \right\|_2 \right)$$
(22)

Coming to the second term of (18), we have:

$$\left\| N^* U\Lambda^{-1} U^\top L^* \right\|_\infty$$
$$= \max_{i,j} \boldsymbol{e}_i{}^\top N^* U\Lambda^{-1} U^\top L^* \boldsymbol{e}_j$$
$$\leq \max_i \left\| \boldsymbol{e}_i{}^\top N^* U \right\|_2 \left\| \Lambda^{-1} U^\top U^* \Sigma^* \right\|_2 \left\| (U^*)^\top \boldsymbol{e}_j \right\|_2$$
$$\leq \sqrt{n} \|N^*\|_\infty \left\| \Lambda^{-1} U^\top U^* \Sigma^* \right\|_2 \frac{\mu\sqrt{r}}{\sqrt{n}} = \mu\sqrt{r} \|N^*\|_\infty \left\| U\Lambda^{-1} U^\top U^* \Sigma^* (U^*)^\top \right\|_2 .$$
(23)

Using an expansion along the lines of (20), we see that

$$\left\| U\Lambda^{-1} U^\top U^* \Sigma^* (U^*)^\top \right\|_2 \leq 1 + \frac{\left\| N^* + E^{(t)} \right\|_2}{|\lambda_k|} \leq 1 + \frac{\|N^*\|_2 + \left\| E^{(t)} \right\|_2}{(1 - 17\mu^2 r \cdot \alpha) \left| \sigma_k^* \right|}$$

$$\leq 2 + \frac{\left\| E^{(t)} \right\|_2}{(1 - 17\mu^2 r\alpha) \left| \sigma_k^* \right|}.$$

Plugging this in (23) gives us

$$\left\|N^* U \Lambda^{-1} U^\top L^*\right\|_\infty \le 3\mu\sqrt{r}\left\|N^*\right\|_\infty. \tag{24}$$

A similar argument as in (23) gives us the following bound on the last term in (18):

$$\left\|N^* U \Lambda^{-1} U^\top N^*\right\|_\infty \le \frac{n\left\|N^*\right\|_\infty^2}{|\lambda_k|} \le \left\|N^*\right\|_\infty. \tag{25}$$

Plugging (22), (24) and (25), we obtain:

$$\left\|(L^* + N^*)\, U \Lambda^{-1} U^\top\, (L^* + N^*) - L^*\right\|_\infty$$
$$\le \frac{\mu^2 r}{n}\left(|\sigma_{k+1}^*| + 7\left\|N^*\right\|_2 + 7\left\|E^{(t)}\right\|_2 + \frac{7n}{\sqrt{r}}\left\|N^*\right\|_\infty\right). \tag{26}$$

Next, we analyze $\sum_{p+q\ge 1}\left\|(E^{(t)})^p(L^* + N^*)U\Lambda^{-(p+q+1)}U^\top(L^* + N^*)(E^{(t)})^q\right\|_\infty$. This can again be bounded by four quantities:

$$\left\|(E^{(t)})^p(L^* + N^*)U\Lambda^{-(p+q+1)}U^\top(L^* + N^*)(E^{(t)})^q\right\|_\infty$$
$$\le \left\|(E^{(t)})^p L^* U\Lambda^{-(p+q+1)}U^\top L^*(E^{(t)})^q\right\|_\infty + \left\|(E^{(t)})^p N^* U\Lambda^{-(p+q+1)}U^\top L^*(E^{(t)})^q\right\|_\infty \tag{27}$$
$$+ \left\|(E^{(t)})^p L^* U\Lambda^{-(p+q+1)}U^\top N^*(E^{(t)})^q\right\|_\infty + \left\|(E^{(t)})^p N^* U\Lambda^{-(p+q+1)}U^\top N^*(E^{(t)})^q\right\|_\infty. \tag{28}$$

We bound the first term above:

$$\left\|(E^{(t)})^p L^* U\Lambda^{-(p+q+1)}U^\top L^*(E^{(t)})^q\right\|_\infty$$
$$= \max_{ij}\boldsymbol{e}_i^\top\left((E^{(t)})^p L^* U\Lambda^{-(p+q+1)}U^\top L^*(E^{(t)})^q\right)\boldsymbol{e}_j,$$
$$\le \max_{ij}\left\|\boldsymbol{e}_i^\top(E^{(t)})^p U^*\right\|_2\left\|\boldsymbol{e}_j^\top(E^{(t)})^q U^*\right\|_2\left\|L^* U\Lambda^{-(p+q+1)}U^\top L^*\right\|_2,$$
$$\stackrel{(\zeta_1)}{\le}\frac{\mu^2 r}{n}\left(\alpha n\left\|E^{(t)}\right\|_\infty\right)^p\left(\alpha n\left\|E^{(t)}\right\|_\infty\right)^q\left\|L^* U\Lambda^{-(p+q+1)}U^\top L^*\right\|_2, \tag{29}$$

where $(\zeta_1)$ follows from Lemma 5 and the incoherence of $L^*$. Now, similar to (20), we have:

$$\left\|L^* U\Lambda^{-(p+q+1)}U^\top L^*\right\|_2$$
$$= \left\|U\Lambda^{-(p+q-1)}U^\top - \left(N^* + E^{(t)}\right)U\Lambda^{-(p+q)}U^\top - U\Lambda^{-(p+q)}U^\top\left(N^* + E^{(t)}\right)\right.$$
$$\left. + \left(N^* + E^{(t)}\right)U\Lambda^{-(p+q+1)}U^\top\left(N^* + E^{(t)}\right)\right\|_2,$$
$$\le \|\Lambda^{-(p+q-1)}\|_2 + 2\|N^* + E^{(t)}\|_2\|\Lambda^{-(p+q)}\|_2 + \|N^* + E^{(t)}\|_2^2\|\Lambda^{-(p+q+1)}\|_2,$$
$$\le |\lambda_k|^{-(p+q-1)}\left(1 + 2\frac{\|N^* + E^{(t)}\|_2}{|\lambda_k|} + \frac{\|N^* + E^{(t)}\|_2^2}{\lambda_k^2}\right),$$
$$= |\lambda_k|^{-(p+q-1)}\left(1 + \frac{\|N^* + E^{(t)}\|_2}{|\lambda_k|}\right)^2,$$
$$\le |\lambda_k|^{-(p+q-1)}\left(1 + \frac{\|N^*\|_2 + \left\|E^{(t)}\right\|_2}{|\lambda_k|}\right)^2,$$
$$\stackrel{(\zeta_1)}{\le}|\lambda_k|^{-(p+q-1)}\left(1 + \frac{\|N^*\|_2 + 17\mu^2 r\alpha\,|\sigma_k^*|}{(1 - 17\mu^2 r\alpha)\,|\sigma_k^*|}\right)^2 \le 2\,|\lambda_k|^{-(p+q-1)}, \tag{30}$$

where $(\zeta_1)$ follows from Lemma 12 and the bound on $\|N^*\|_\infty$.

Using (29), (30), we have:

$$\left\| (E^{(t)})^p L^* U \Lambda^{-(p+q+1)} U^\top L^* (E^{(t)})^q \right\|_\infty \leq 2\alpha\mu^2 r \left\| E^{(t)} \right\|_\infty \left( \frac{\alpha n \left\| E^{(t)} \right\|_\infty}{|\lambda_k|} \right)^{p+q-1}. \quad (31)$$

Coming to the second term of (28), we have

$$\left\| (E^{(t)})^p N^* U \Lambda^{-(p+q+1)} U^\top L^* (E^{(t)})^q \right\|_\infty$$

$$= \max_{i,j} \boldsymbol{e}_i^\top \left( (E^{(t)})^p N^* U \Lambda^{-(p+q+1)} U^\top L^* (E^{(t)})^q \right) \boldsymbol{e}_j,$$

$$\leq \max_{ij} \left\| \boldsymbol{e}_i^\top (E^{(t)})^p N^* U \right\|_2 \left\| \boldsymbol{e}_j^\top (E^{(t)})^q U^* \right\|_2 \left\| \Lambda^{-(p+q+1)} U^\top L^* \right\|_2$$

$$\overset{(\zeta_1)}{\leq} \frac{\mu\sqrt{r}}{\sqrt{n}} \| N^* U \|_\infty \left( \alpha n \left\| E^{(t)} \right\|_\infty \right)^p \left( \alpha n \left\| E^{(t)} \right\|_\infty \right)^q \left\| U \Lambda^{-(p+q+1)} U^\top L^* \right\|_2$$

$$\leq \mu\sqrt{r} \| N^* \|_\infty \left( \alpha n \left\| E^{(t)} \right\|_\infty \right)^{p+q} \left\| U \Lambda^{-(p+q+1)} U^\top L^* \right\|_2, \quad (32)$$

where $(\zeta_1)$ follows from Lemma 5 and incoherence of $U^*$. Proceeding along the lines of (30), we obtain:

$$\left\| U \Lambda^{-(p+q+1)} U^\top L^* \right\|_2 \leq |\lambda_k|^{-(p+q)} \left( 1 + \frac{\| N^* \|_2 + \| E^{(t)} \|_2}{|\lambda_k|} \right) \leq 2 |\lambda_k|^{-(p+q)}.$$

Plugging the above in (32) gives us

$$\left\| (E^{(t)})^p N^* U \Lambda^{-(p+q+1)} U^\top L^* (E^{(t)})^q \right\|_\infty \leq 2\mu\sqrt{r} \left( \frac{\alpha n \left\| E^{(t)} \right\|_\infty}{|\lambda_k|} \right)^{p+q} \| N^* \|_\infty. \quad (33)$$

A similar argument as in (32) gives us

$$\left\| (E^{(t)})^p N^* U \Lambda^{-(p+q+1)} U^\top N^* (E^{(t)})^q \right\|_\infty \leq \frac{n \| N^* \|_\infty}{|\lambda_k|} \left( \frac{\alpha n \left\| E^{(t)} \right\|_\infty}{|\lambda_k|} \right)^{p+q} \| N^* \|_\infty.$$

Plugging the above inequality along with (31) and (33) into (28) gives us:

$$\left\| (E^{(t)})^p (L^* + N^*) U \Lambda^{-(p+q+1)} U^\top (L^* + N^*) (E^{(t)})^q \right\|_\infty$$

$$\leq 2\mu^2 r \left( \alpha \left\| E^{(t)} \right\|_\infty + \frac{\| N^* \|_\infty}{\sqrt{r}} \right) \left( \frac{\alpha n \left\| E^{(t)} \right\|_\infty}{|\lambda_k|} \right)^{p+q-1}.$$

Using the above bound, and the assumption on $\left\| E^{(t)} \right\|_\infty$:

$$\left\| E^{(t)} \right\|_\infty \leq \frac{8\mu^2 r}{n} \left( |\sigma_{k+1}^*| + \left( \frac{1}{2} \right)^{t-1} |\sigma_k^*| + 7 \| N^* \|_2 + \frac{8n}{\sqrt{r}} \| N^* \|_\infty \right) \leq \frac{17\mu^2 r}{n} |\sigma_k^*|,$$

we have:

$$\sum_{p+q \geq 1} \left\| (E^{(t)})^p (L^* + N^*) U \Lambda^{-(p+q+1)} U^\top (L^* + N^*) (E^{(t)})^q \right\|_\infty \quad (34)$$

$$\leq 2\mu^2 r \left( \alpha \left\| E^{(t)} \right\|_\infty + \frac{\| N^* \|_\infty}{\sqrt{r}} \right) \sum_{p+q \geq 1} \left( \frac{\alpha n \left\| E^{(t)} \right\|_\infty}{|\lambda_k|} \right)^{p+q-1}$$

$$\leq 2\mu^2 r \left( \alpha \left\| E^{(t)} \right\|_\infty + \frac{\| N^* \|_\infty}{\sqrt{r}} \right) \left( \frac{1}{1 - \frac{17\mu^2 \alpha r}{1 - 17\mu^2 \alpha \cdot r}} \right)^2$$

$$\leq 2\mu^2 r \left( \alpha \left\| E^{(t)} \right\|_\infty + \frac{\| N^* \|_\infty}{\sqrt{r}} \right) \left( \frac{1}{1 - 34\mu^2 r\alpha} \right)^2$$

$$\leq 4\mu^2 r \left( \alpha \left\| E^{(t)} \right\|_\infty + \frac{\| N^* \|_\infty}{\sqrt{r}} \right). \quad (35)$$

Combining (17), (26), (35), we have:

$$
\left\| U\Lambda U^\top - L^* \right\|_\infty \overset{(\zeta_1)}{\leq} \frac{\mu^2 r}{n} \left( |\sigma^*_{k+1}| + 7\left\| N^* \right\|_2 + 11n\alpha \left\| E^{(t)} \right\|_\infty + \frac{11n}{\sqrt{r}} \left\| N^* \right\|_\infty \right)
$$

$$
\overset{(\zeta_2)}{\leq} \frac{2\mu^2 r}{n} \left( |\sigma^*_{k+1}| + \left(\frac{1}{2}\right)^t |\sigma^*_k| + 7\left\| N^* \right\|_2 + \frac{8n}{\sqrt{r}} \left\| N^* \right\|_\infty \right),
$$

where $(\zeta_1)$ follows from Lemma 4, and $(\zeta_2)$ follows from the assumption on $\left\| E^{(t)} \right\|_\infty$. □

We used the following technical lemma in the proof of Lemma 11.

**Lemma 12.** *Assume the notation of Lemma 11. Suppose further that*

1. $\left\| E^{(t)} \right\|_\infty \leq \frac{8\mu^2 r}{n} \left( |\sigma^*_{k+1}| + \left(\frac{1}{2}\right)^{t-1} |\sigma^*_k| + 7\left\| N^* \right\|_2 + \frac{8n}{\sqrt{r}} \left\| N^* \right\|_\infty \right)$, *and*

2. $\mathrm{Supp}\left( E^{(t)} \right) \subseteq \mathrm{Supp}\left( S^* \right).$

*Then we have:*

$$
\left\| E^{(t)} \right\|_2 \leq 17\mu^2 r\alpha |\sigma^*_k|, \quad |\lambda_k| \geq |\sigma^*_k|(1 - 17\mu^2 r\alpha), \quad \text{and} \quad |\lambda_{k+1}| \leq |\sigma^*_{k+1}| + \left\| E^{(t)} \right\|_2.
$$

*Proof.* Using Lemmas 4 and 2, we have:

$$
|\lambda_i - \sigma^*_i| \leq \left\| E^{(t)} \right\|_2 \leq \alpha n \left\| E^{(t)} \right\|_\infty.
$$

Using the bound on $\left\| E^{(t)} \right\|_\infty$ and recalling the assumption that

$$
\| N^* \|_\infty \leq \frac{|\sigma^*_r|}{100}
$$

finishes the proof. □

The following lemma bounds the support of $E^{(t+1)}$ and $\left\| E^{(t+1)} \right\|_\infty$, using an assumption on $\left\| L^{(t+1)} - L^* \right\|_\infty$.

**Lemma 13.** *Assume the notation of Lemma 11. Suppose*

$$
\left\| L^{(t+1)} - L^* \right\|_\infty \leq \frac{2\mu^2 r}{n} \left( |\sigma^*_{k+1}| + \left(\frac{1}{2}\right)^t |\sigma^*_k| + 7\left\| N^* \right\|_2 + \frac{8n}{\sqrt{r}} \left\| N^* \right\|_\infty \right).
$$

*Then, we have:*

1. $\mathrm{Supp}\left( E^{(t+1)} \right) \subseteq \mathrm{Supp}\left( S^* \right).$

2. $\left\| E^{(t+1)} \right\|_\infty \leq \frac{7\mu^2 r}{n} \left( |\sigma^*_{k+1}| + \left(\frac{1}{2}\right)^t |\sigma^*_k| + 7\left\| N^* \right\|_2 + \frac{8n}{\sqrt{r}} \left\| N^* \right\|_\infty \right)$, *and*

*Proof.* We first prove the first conclusion. Recall that,

$$
S^{(t+1)} = H_\zeta(M - L^{(t+1)}) = H_\zeta(L^* - L^{(t+1)} + N^* + S^*),
$$

where $\zeta = \frac{4\mu^2 r}{n} \left( |\lambda_{k+1}| + \left(\frac{1}{2}\right)^t |\lambda_k| \right)$ is as defined in Algorithm 1 and $\lambda_1, \cdots, \lambda_n$ are the eigenvalues of $M - S^{(t)}$ such that $|\lambda_1| \geq \cdots \geq |\lambda_n|$.

If $S^*_{ij} = 0$ then $E^{(t+1)}_{ij} = \mathbb{1}_{\left\{ \left| L^*_{ij} - L^{(t+1)}_{ij} + N^*_{ij} \right| > \zeta \right\}} \cdot (L^*_{ij} - L^{(t+1)}_{ij} + N^*_{ij})$. The first part of the lemma

now follows by using the assumption that $\left\| L^{(t+1)} - L^* \right\|_\infty \leq \frac{2\mu^2 r}{n} \left( |\sigma^*_{k+1}| + \left(\frac{1}{2}\right)^t |\sigma^*_k| \right) \overset{(\zeta_1)}{\leq}$ $\frac{9\mu^2 r}{4n} \left( |\lambda^*_{k+1}| + \left(\frac{1}{2}\right)^t |\lambda^*_k| \right) = \zeta$, where $(\zeta_1)$ follows from Lemma 6, and the bound on $\| N^* \|_\infty$.

We now prove the second conclusion. We consider the following two cases:

1. $\left|M_{ij} - L_{ij}^{(t+1)}\right| > \zeta$: Here, $S_{ij}^{(t+1)} = S_{ij}^* + L_{ij}^* - L_{ij}^{(t+1)} + N_{ij}^*$. Hence, $|S_{ij}^{(t+1)} - S_{ij}^*| \leq$
   $|L_{ij}^* - L_{ij}^{(t+1)}| + |N_{ij}^*| \leq \frac{2\mu^2 r}{n}\left(|\sigma_{k+1}^*| + \left(\frac{1}{2}\right)^t |\sigma_k^*|\right) + \|N^*\|_\infty$.

2. $\left|M_{ij} - L_{ij}^{(t+1)}\right| \leq \zeta$: In this case, $S_{ij}^{(t+1)} = 0$ and $\left|S_{ij}^* + L_{ij}^* - L_{ij}^{(t+1)} + N_{ij}^*\right| \leq \zeta$. So
   we have, $\left|E_{ij}^{(t+1)}\right| = |S_{ij}^*| \leq \zeta + \left|L_{ij}^* - L_{ij}^{(t+1)}\right| + |N_{ij}^*| \leq \frac{7\mu^2 r}{n}\left(|\sigma_{k+1}^*| + \left(\frac{1}{2}\right)^t |\sigma_k^*|\right) +$
   $\|N^*\|_\infty$. The last inequality above follows from Lemma 6.

This proves the lemma. $\qquad\qquad\qquad\qquad\qquad\qquad\qquad\qquad\qquad\qquad\qquad\qquad\square$

The following lemma is a generalization of Lemma 1.

**Lemma 14.** *Let $L^*, S^*, N^*$ be symmetric and satisfy the assumptions of Theorem 2 and let $M^{(t)}$ and $L^{(t)}$ be the $t^{th}$ iterates of the $k^{th}$ stage of Algorithm 1. Let $\sigma_1^*, \ldots, \sigma_n^*$ be the eigenvalues of $L^*$, s.t., $|\sigma_1^*| \geq \cdots \geq |\sigma_r^*|$. Then, the following holds:*

$$\left\|L^{(t+1)} - L^*\right\|_\infty \leq \frac{2\mu^2 r}{n}\left(|\sigma_{k+1}^*| + \left(\frac{1}{2}\right)^t |\sigma_k^*| + 7\|N^*\|_2 + \frac{8n}{\sqrt{r}}\|N^*\|_\infty\right),$$

$$\left\|E^{(t+1)}\right\|_\infty = \left\|S^* - S^{(t+1)}\right\|_\infty \leq \frac{8\mu^2 r}{n}\left(|\sigma_{k+1}^*| + \left(\frac{1}{2}\right)^{t-1} |\sigma_k^*| + 7\|N^*\|_2 + \frac{8n}{\sqrt{r}}\|N^*\|_\infty\right), \text{ and}$$

$$\text{Supp}\left(E^{(t+1)}\right) \subseteq \text{Supp}\left(S^*\right).$$

*Moreover, the outputs $\widehat{L}$ and $\widehat{S}$ of Algorithm 1 satisfy:*

$$\left\|\widehat{L} - L^*\right\|_F \leq \epsilon + 2\mu^2 r\left(7\|N^*\|_2 + \frac{8n}{\sqrt{r}}\|N^*\|_\infty\right),$$

$$\left\|\widehat{S} - S^*\right\|_\infty \leq \frac{\epsilon}{n} + \frac{8\mu^2 r}{n}\left(7\|N^*\|_2 + \frac{8n}{\sqrt{r}}\|N^*\|_\infty\right), \text{ and}$$

$$\text{Supp}\left(\widehat{S}\right) \subseteq \text{Supp}\left(S^*\right).$$

*Proof.* Recall that in the $k^{\text{th}}$ stage, the update $L^{(t+1)}$ is given by: $L^{(t+1)} = P_k(M - S^{(t)})$ and $S^{(t+1)}$ is given by: $S^{(t+1)} = H_\zeta(M - L^{(t+1)})$. Also, recall that $E^{(t)} := S^* - S^{(t)}$ and $E^{(t+1)} := S^* - S^{(t+1)}$.

We prove the lemma by induction on both $k$ and $t$. For the base case ($k = 1$ and $t = -1$), we first note that the first inequality on $\left\|L^{(0)} - L^*\right\|_\infty$ is trivially satisfied. Due to the thresholding step (step 3 in Algorithm 1) and the incoherence assumption on $L^*$, we have:

$$\left\|E^{(0)}\right\|_\infty \leq \frac{8\mu^2 r}{n}\left(\sigma_2^* + 2\sigma_1^*\right), \text{ and}$$

$$\text{Supp}\left(E^{(0)}\right) \subseteq \text{Supp}\left(S^*\right).$$

So the base case of induction is satisfied.

We first do the inductive step over $t$ (for a fixed $k$). By inductive hypothesis we assume that: a) $\left\|E^{(t)}\right\|_\infty \leq \frac{8\mu^2 r}{n}\left(|\sigma_{k+1}^*| + \left(\frac{1}{2}\right)^{t-1} |\sigma_k^*| + 7\|N^*\|_2 + \frac{8n}{\sqrt{r}}\|N^*\|_\infty\right)$, b) $\text{Supp}\left(E^{(t)}\right) \subseteq \text{Supp}\left(S^*\right)$. Then by Lemma 11, we have:

$$\left\|L^{(t+1)} - L^*\right\|_\infty \leq \frac{2\mu^2 r}{n}\left(|\sigma_{k+1}^*| + \left(\frac{1}{2}\right)^t |\sigma_k^*| + 7\|N^*\|_2 + \frac{8n}{\sqrt{r}}\|N^*\|_\infty\right).$$

Lemma 13 now tells us that

1. $\left\|E^{(t+1)}\right\|_\infty \leq \frac{7\mu^2 r}{n}\left(|\sigma_{k+1}^*| + \left(\frac{1}{2}\right)^t |\sigma_k^*| + 7\|N^*\|_2 + \frac{8n}{\sqrt{r}}\|N^*\|_\infty\right)$, and

2. $\mathrm{Supp}\left(E^{(t+1)}\right) \subseteq \mathrm{Supp}\left(S^*\right)$.

This finishes the induction over $t$. Note that we show a stronger bound than necessary on $\left\|E^{(t+1)}\right\|_\infty$.

We now do the induction over $k$. Suppose the hypothesis holds for stage $k$. Let $T$ denote the number of iterations in each stage. We first obtain a lower bound on $T$. Since

$$\left\|M - S^{(0)}\right\|_2 \geq \|L^* + N^*\|_2 - \left\|E^{(0)}\right\|_2 \geq |\sigma_1^*| - \alpha n \left\|E^{(0)}\right\|_\infty \geq \frac{3}{4}|\sigma_1^*|,$$

we see that $T \geq 10\log\left(3\mu^2 r\,|\sigma_1^*|\,/\epsilon\right)$. So, at the end of stage $k$, we have:

1. $\left\|E^{(T)}\right\|_\infty \leq \frac{7\mu^2 r}{n}\left(|\sigma_{k+1}^*| + \left(\frac{1}{2}\right)^T |\sigma_k^*| + 7\,\|N^*\|_2 + \frac{8n}{\sqrt{r}}\,\|N^*\|_\infty\right) \leq \frac{7\mu^2 r |\sigma_{k+1}^*|}{n} + \frac{\epsilon}{10n}$, and

2. $\mathrm{Supp}\left(E^{(T)}\right) \subseteq \mathrm{Supp}\left(S^*\right)$.

Lemmas 4 and 2 tell us that $\left|\sigma_{k+1}\left(M - S^{(T)}\right) - |\sigma_{k+1}^*|\right| \leq \left\|E^{(T)}\right\|_2 \leq \alpha\left(7\mu^2 r\,|\sigma_{k+1}^*| + \epsilon\right)$. We will now consider two cases:

1. **Algorithm 1 terminates:** This means that $\beta\sigma_{k+1}\left(M - S^{(T)}\right) < \frac{\epsilon}{2n}$ which then implies that $|\sigma_{k+1}^*| < \frac{\epsilon}{6\mu^2 r}$. So we have:

$$\left\|\widehat{L} - L^*\right\|_\infty = \left\|L^{(T)} - L^*\right\|_\infty \leq \frac{2\mu^2 r}{n}\left(|\sigma_{k+1}^*| + \left(\frac{1}{2}\right)^T |\sigma_k^*| + 7\,\|N^*\|_2 + \frac{8n}{\sqrt{r}}\,\|N^*\|_\infty\right)$$
$$\leq \frac{\epsilon}{5n} + \frac{2\mu^2 r}{n}\left(7\,\|N^*\|_2 + \frac{8n}{\sqrt{r}}\,\|N^*\|_\infty\right).$$

This proves the statement about $\widehat{L}$. A similar argument proves the claim on $\left\|\widehat{S} - S^*\right\|_\infty$. The claim on $\mathrm{Supp}\left(\widehat{S}\right)$ follows since $\mathrm{Supp}\left(E^{(T)}\right) \subseteq \mathrm{Supp}\left(S^*\right)$.

2. **Algorithm 1 continues to stage $(k+1)$:** This means that $\beta\sigma_{k+1}\left(L^{(T)}\right) \geq \frac{\epsilon}{2n}$ which then implies that $|\sigma_{k+1}^*| > \frac{\epsilon}{8\mu^2 r}$. So we have:

$$\left\|E^{(T)}\right\|_\infty \leq \frac{7\mu^2 r}{n}\left(|\sigma_{k+1}^*| + \left(\frac{1}{2}\right)^T |\sigma_k^*| + 7\,\|N^*\|_2 + \frac{8n}{\sqrt{r}}\,\|N^*\|_\infty\right)$$
$$\leq \frac{7\mu^2 r}{n}\left(|\sigma_{k+1}^*| + \frac{\epsilon}{10\mu^2 rn} + 7\,\|N^*\|_2 + \frac{8n}{\sqrt{r}}\,\|N^*\|_\infty\right)$$
$$\leq \frac{7\mu^2 r}{n}\left(|\sigma_{k+1}^*| + \frac{8\,|\sigma_{k+1}^*|}{10n} + 7\,\|N^*\|_2 + \frac{8n}{\sqrt{r}}\,\|N^*\|_\infty\right)$$
$$\leq \frac{8\mu^2 r}{n}\left(|\sigma_{k+2}^*| + 2\,|\sigma_{k+1}^*| + 7\,\|N^*\|_2 + \frac{8n}{\sqrt{r}}\,\|N^*\|_\infty\right).$$

Similarly for $\left\|L^{(T)} - L^*\right\|_\infty$.

This finishes the proof. $\qquad\square$

*Proof of Theorem 2.* Using Lemma 14, it suffices to show that the general case can be reduced to the case of symmetric matrices. We will now outline this reduction.

Recall that we are given an $m \times n$ matrix $M = L^* + N^* + S^*$ where $L^*$ is the true low-rank matrix, $N^*$ dense corruption matrix and $S^*$ the sparse error matrix. Wlog, let $m \leq n$ and suppose

Figure 5: (a): Variation of the maximum rank of the intermediate low-rank solutions of IALM with rank. (b): Variation of the maximum rank of the intermediate low-rank solutions of IALM with incoherence. (c): Rank of the intermediate iterates of IALM for a particular run with $n = 2000, r = 10, \alpha = 100/n, \mu = 3$. Note that while the rank of the final output is 10, intermediate iterates have rank as high as $800$.

$\beta m \leq n < (\beta + 1)m$, for some $\beta \geq 1$. We then consider the symmetric matrices

$$\widetilde{M} = \underbrace{\begin{bmatrix} 0 & 0 & M \\ \vdots & \cdots & \vdots & \vdots \\ 0 & 0 & M \\ M^\top & \cdots M^\top & 0 \end{bmatrix}}_{\beta \text{ times}}, \widetilde{L} = \underbrace{\begin{bmatrix} 0 & 0 & L^* \\ \vdots & \cdots & \vdots & \vdots \\ 0 & 0 & L^* \\ (L^*)^\top & \cdots (L^*)^\top & 0 \end{bmatrix}}_{\beta \text{ times}},$$

$$\widetilde{N} = \underbrace{\begin{bmatrix} 0 & 0 & L^* \\ \vdots & \cdots & \vdots & \vdots \\ 0 & 0 & L^* \\ (N^*)^\top & \cdots (N^*)^\top & 0 \end{bmatrix}}_{\beta \text{ times}}, \tag{36}$$

and $\widetilde{S} = \widetilde{M} - \widetilde{L}$. A simple calculation shows that $\widetilde{L}$ is incoherent with parameter $\sqrt{3}\mu$, $\widetilde{N}$ satisfies the assumption of Theorem 2 and $\widetilde{S}$ satisfies the sparsity condition (S1) with parameter $\frac{\alpha}{\sqrt{2}}$. Moreover the iterates of AltProj with input $\widetilde{M}$ have similar expressions as in (36) in terms of the corresponding iterates with input $M$. This means that it suffices to obtain the same guarantees for Algorithm 1 for the symmetric case. Lemma 14 does precisely this, proving the theorem. $\square$

## C  Additional experimental results

**Synthetic datasets:**  Extending Figure 2, the plots in Figure 5 illustrate the point that soft thresholding, i.e., the convex relaxation approach, leads to intermediate solutions with high ranks. Figures 5 (a)-(b) show the variation of the maximum rank of the intermediate low-rank solutions of IALM with rank and incoherence respectively; the results are averaged over 5 runs of the algorithm; we note that as the problem becomes harder, the maximum intermediate rank via soft thresholding (convex approach) increases, and this leads to higher running times. As an example of this phenomenon, Figure 5 (c) shows the rank of the intermediate iterates of IALM for a particular run with $n = 2000, r = 10, \alpha = 100/n, \mu = 3$; here, while the rank of the final output is 10, intermediate iterates have a rank as high as $800$. We run our synthetic simulations on a machine with Intel Dual 8-core Xeon (E5-2650) 2.0GHz CPU with 192GB RAM.

**Real-world datasets:**  We provide some additional results concerning foreground-background separation in videos [5]. We compare NcRPCA with IALM, and also with the low-rank solution

Figure 6: Foreground-background separation in the *Shopping Mall* video. (a): Original frame in the video given as a part of the input to NcRPCA and IALM. (b): Corresponding frame from the best rank-20 approximation obtained using vanilla PCA; time taken for computing the low-rank approximation is $8.8s$. (c): Corresponding frame from the low-rank part obtained using NcRPCA; time taken by NcRPCA to compute the low-rank and sparse solutions is $292.1s$. (d): Corresponding frame from the sparse part obtained using NcRPCA. (e): Corresponding frame from the low-rank part obtained using IALM; time taken by IALM to compute the low-rank and sparse solutions is $783.4s$. (f): Corresponding frame from the sparse part obtained using IALM.

obtained using vanilla PCA; we report the solutions obtained by NcRPCA and IALM methods for decomposing $M$ into $L + S$ up to a relative error ($\|M - L - S\|_F / \|M\|_F$) of $10^{-3}$. We report the rank and the sparsity of the solutions obtained by the two methods along with the computational time. As mentioned before, the observed matrix $M$ is formed by vectorizing each frame and stacking them column-wise. For illustration purposes, we arbitrarily select one of the original frames in the sequence of image frames obtained from the video, i.e., one of the columns of $M$, and the corresponding columns in $L$ and $S$ obtained using NcRPCA and IALM. We run our real data experiments on a machine with Intel Dual 8-core Xeon (E5-2650) 2.0GHz CPU with 192GB RAM.

*Shopping Mall dataset:* Figure 6 shows the comparison of NcRPCA and IALM on the "Shopping Mall" dataset which has 1286 frames at a resolution of $256 \times 320$. NcRPCA achieves a solution of better visual quality (for example, unlike NcRPCA, notice the artifact of the low-rank solution from IALM in the top right corner of the image where the person is walking over the reflection of a light source; also notice the shadows of people in the low-rank part obtained by IALM which are not present in the low-rank solution obtained by NcRPCA), in $292.1s$, compared to IALM, which takes $783.4s$ until convergence. NcRPCA obtains a rank 20 solution for $L$ with $\|S\|_0 = 95411896$ whereas IALM obtains a rank 286 solution for $L$ with $\|S\|_0 = 86253965$.

*Curtain dataset:* We illustrate our recovery on one of the frames (frame 2773) wherein a person enters a room with a curtain on the background. Figure 7 shows the comparison of NcRPCA and IALM on the "Curtain" dataset which has 2964 frames at a resolution of $160 \times 128$. NcRPCA achieves a solution, in $39.5s$, which is of similar visual quality to that of IALM, which takes $989.0s$ until convergence. NcRPCA obtains a rank 1 solution for $L$ with $\|S\|_0 = 53897769$ whereas IALM obtains a rank 701 solution for $L$ with $\|S\|_0 = 42310582$.

Figure 7: Foreground-background separation in the *Curtain* video. (a): Original image frame in the video given as a part of the input to NcRPCA and IALM. (b): Corresponding frame from the best rank-10 approximation obtained using vanilla PCA; time taken for computing the low-rank approximation is $2.8s$. (c): Corresponding frame from the low-rank part obtained using NcRPCA; time taken by NcRPCA to compute the low-rank and sparse solutions is $39.5s$. (d): Corresponding frame from the sparse part obtained using NcRPCA. (e): Corresponding frame from the low-rank part obtained using IALM; time taken by IALM to compute the low-rank and sparse solutions is $989.0s$. (f): Corresponding frame from the sparse part obtained using IALM.

## Footnotes

[5]The datasets are available at `http://perception.i2r.a-star.edu.sg/bk_model/bk_index.html`