[Reviews · NeurIPS 2014]

Submitted by Assigned_Reviewer_1

The authors present a method for Robust PCA i.e. decomposing a matrix into the sum of a low-rank component and a sparse component. The method works by setting the current iterate entires to the corresponding entries of its rank-k projection whenever the difference does not exceed a threshold. The threshold is then decreased and the process is repeated until converge. This inner loop is repeated for k=1, ..., r.

The authors provide strong theoretical and empirical evidence that (1) guarantee recovery under certain conditions (incoherence of the underlying low-rank component, uniform sparsity of the sparse component) which have been assumed in previous results, and (2) demonstrate a faster rate of convergence (log(1/\epsilon) vs. 1/\epsilon for previous methods) akin to that of standard PCA and (3) faster complexity per iteration (O(r^2mn) vs. O(mn^2) for previous methods) that is almost that of standard PCA (up to a factor of r).

This work is an example of how iteratively optimizing a non-convex objective can be better (in terms of speed and accuracy) than optimizing the convex relaxation. It is of high quality and is very clearly written. The method is novel to my knowledge, and is a very significant contribution due to its superior convergence/speed, as well as its recovery guarantees.

Some questions:

- Selection of \beta: you recommend an aggressive setting (\sqrt{n}) of \beta in practice, while the theoretical results rely on \beta = 2*\mu^2*r/n. Is there a relationship between these two (there need not be but I am curious)?

- The symmetry assumption on the noise matrix seems restrictive. Does the same logic as in Eq.(3) work for reducing to the symmetric case when there is noise?

- Conclusion/Discussion?

Minor corrections:

Line 209: "Let L* = ..."
Line 210-211: Typically the SVD L = USV* assumes U, V are unitary, so it's unclear what you mean when you assume ||(U^*)^i||_2 <= \mu * \sqrt{r/m} and likewise for ||(V^*)^i||_2
Line 241: the bound on the number of iterations here has no dependence on m (in Theorem 1, there was a dependence). Is this intended, and if so, what accounts for this difference in the noisy case?
Line 420: qualtiy --> quality

Summary: This paper represents a significant contribution to a well-known problem of Robust PCA by providing a simple iterative method that has theoretical guarantees as as enjoys convergence/complexity that is almost that of regular PCA. The authors also demonstrate the superiority of this method on simulated and real applications.

Submitted by Assigned_Reviewer_23

The paper proposes a novel and practical approach to robust PCA whose task is to recover a low-rank matrix, that is corrupted with sparse perturbations. The suggested method is non-convex and alternates between projections onto the set of low-rank and sparse matrices. The authors are able to prove recovery under tight conditions with favourable runtime complexity compared to state-of-the-art. The improved computational complexity allows scalability and enables much faster computation as convincingly demonstrated on the example problem of foreground-background separation of video sequences. The key idea is an intermediate and iteratively applied denoising step that suppresses sparse errors by discarding matrix elements with large approximation errors. In a number of synthetic and real-world examples the authors demonstrate the benefits of their method.

The paper is well written, clearly structured and seems technically sound (though I didn't check in full detail). Unfortunately, a conclusion and a discussion on potential future work, open questions, existing shortcomings are missing and should be included for the final version.

Due to the improved runtime complexity incl. tight guarantees, the proposed method seems likely to be adopted in real-world applications and to trigger follow-up work within the NIPS community.

Summary: Innovative, mathematically sound and practical approach to robust PCA with improved runtime complexity.

Submitted by Assigned_Reviewer_44

This paper is about developing non-convex counter part to Robust PCA. Robust PCA is a popular convex optimization scheme for decomposing a low rank matrix which is corrupted with sparse perturbations into its sparse and low rank components. However, the current methods based on convex optimization are computationally prohibitive as they require either matrix inversion and or full singular value decomposition. This paper develops a non-convex approach to this problem. The authors' simulations indicate that this non-convex algorithm can be much faster. The authors also provide some theoretical analysis for their algorithm.

I think the subject of this paper is very timely and well suited for NIPS. Overall I find the paper to be interesting and well written.
The technical issues I had raised have been resolved (for the most part) in the authors' response. There are still some issues with authors' claim of matching convex schemes (which I've explained below). However, I trust the authors to address this issue in the final version of the paper. Regardless of this issue I think this paper is a must accept at NIPS and I have updated my score. There also a couple of minor mistakes in the authors' response which I detail below.

- Modified Pros:
-- well written
-- suited for NIPS
-- new algorithm
-- study of non-convex problems (very few such studies in the literature)
-- real data experiments

- Modified Cons:

-- proven in an easier regime compared to convex counter parts
-- need knowledge of problem parameters such as rank and incoherence

Detailed comments regarding the authors' response:

1) ``We match KHZ 11"

The authors do not match the deterministic results of HKZ11. They match it for the case where one is interested in sparsity per row/column. The interesting part about the deterministic result of HKZ is that their conditions can not only give this result but also nearly match the results of CLMW11 for the random models.

2) ``Comparison with CLMW11:"
Even with these added assumptions the author can not at this point match the results of CLMW11. My point is that I think the authors should be careful in their comparison and claims. That being said, I completely agree with the authors that these results can be matched with significantly more evolved proofs.

Summary: This paper is about developing non-convex counter part to Robust PCA (a popular convex optimization scheme for sparse+low rank decomposition). The paper is timely, interesting and well suited for NIPS.
Author Feedback
Author rebuttal: We would like to thank the reviewers for their comments and suggestions.

-Selection of \beta: In experiments we set \beta to be 1/sqrt(n). Our theory assumes worst case errors in the sparse part (e.g., all the elements could be positive and aligned, leading to a spectral norm of d for the sparse error matrix). However, in our experiments, the errors are not adversarial and hence a more aggressive choice of \beta works as well (and makes the algorithm converge faster). Using our theoretical value of \beta will also lead to exact recovery, but with slower convergence.

-Conclusion/discussion section: We omitted conclusion/discussion section due to lack of space. We will be happy to reorganize for the final version.

Reviewer 1:
- Line 210-211: (U^*)^i stands for the i^th row of U^*; the assumption on ||(U^*)^i|| is just the standard incoherence assumption.
- Line 241: Theorem 2 is only for the symmetric case. For the general case, we will have a similar dependence on $m$ as in Theorem 1.

Reviewer 3:
We appreciate your detailed comments and we will incorporate them in revised version. First, we clarify that we match the sparsity level for convex solvers under deterministic sparsity and standard incoherence assumptions. The running time of our method is better than the upper bound on the running time of the convex solvers (there is currently no analysis for convex solvers which improves this upper bound).

We match HKZ11: In Lines 93-96, we claim we match CSPW11. This was a typo and in fact, we match HKZ11 which has the strongest guarantee under deterministic sparsity, which is n/(\mu^2 r) per row/column; see Theorem 1. When r = O(1), this allows for a constant fraction of corrupted entries.

Comparision with CLMW11: CLMW11 assumes random sparsity and additional incoherence: |U V^t|_\infty< \mu \sqrt{r}/n. Our assumption of ||U^(i)||< \mu \sqrt{r/n} only yields |U V^t|_\infty<\mu^2 r/n. Without these additional assumptions, it is impossible to improve the guarantees of HKZ11 setting, which we match.

Knowing incoherence in advance: See lines 358-361. CLMW11 also requires an upper bound on the incoherence parameter to set the regularization parameter. If the incoherence of the underlying matrix is large, a setting of \lambda = 1/\sqrt(n) does not work for RPCA. The same is true for our algorithm; we only need an upper bound on the incoherence.

Other comments:
3-4) The O(.) and o(.) notation only refer to upper bounds (\Omega(.) is used for lower bounds). So, our claim is consistent with the state of the art for convex solvers. We will make sure to state explicitly that we are only talking about upper bounds.

5-6), 12), 14-15), 20) We will mention that our paper matches the results of convex approaches with out the additional assumption on ||UV'||_{\infty}.

8) Please refer our response above about beta

9), 10): Note that we only require low-rank approximation of M^t, and not the actual singular vectors. Low-rank approximation can be performed upto epsilon accuracy in time O(rmn log(1/epsilon)), irrespective of singular values (and their gaps). A popular algorithm here would be power-method or its variants like orthogonal iteration. We will clarify this point in the paper. Also, we can handle the "epsilon" error due to SVD, but ignore it to keep the proofs relatively simple.

11), 25), 28) Typos noted.

17), 24)
- M^0 is just a thresholded version of M. In particular, the uncorrupted elements of M are unchanged and corruptions of very large magnitude are decreased. If the corruptions in M are bounded entrywise, this does not change M. Otherwise, this reduces it to the case where the matrix M has bounded magnitude corruptions.
- Since S* is sparse and its elements are bounded in magnitude (due to thresholding), its largest singular value is much smaller than \lambda (see Lemma 1). So I - S*/\lambda is invertible.

18) There is a typo in (5) and what follows: u'u* should be replaced by (u'u*)^2. But the entire argument still goes through with this modification.

19) S^t = M^t - L* -- please see Lemma 5. Lemma 5 uses the conclusions of Lemma 3 and proves a similar statement for S^t.

21) See Fig 1(a), 1(b) or first point of 1(c). Here mu=1, so 1/sqrt(n) works for RPCA, but here also our method is significantly faster than RPCA.

22-23) Yes, that is a typo. We mean |M_{ij}^t - L_{ij}^t|. \beta should also be chosen to be 2 times larger than what is stated in the theorem.

29) There is a minor notational inconsitency here. The base case is for t = 0 and so S^0 = M^0 - L* = M - L* = S*.

30) We prove Lemma 3 using induction. Our induction hypothesis is over *S^t*, using induction hypothesis and Lemma 7, we obtain bound on L^(t+1)-L^*. We then show that our induction hypothesis (for S^{t+1}) holds for t+1-th step.

31) We threshold the values in M (since we know that the largest entry in L* is at most \mu^2r/n) to ensure ||S*||_infty\leq 2 \mu^2 r/n. See Step 3 of Algorithm 1. Since S^0 = S*, (please refer to our response (29) above) we have the required infinity norm bound

32) Yes, M^t = L* + S^t

33) We will fix/clarify the minor issues in Theorem 2's proof as well.